# One-shot learning and big data with $n = 2$

**Lee H. Dicker**
Rutgers University
Piscataway, NJ
ldicker@stat.rutgers.edu

**Dean P. Foster**
University of Pennsylvania
Philadelphia, PA
dean@foster.net

## Abstract

We model a "one-shot learning" situation, where very few observations $y_1, ..., y_n \in \mathbb{R}$ are available. Associated with each observation $y_i$ is a very high-dimensional vector $x_i \in \mathbb{R}^d$, which provides context for $y_i$ and enables us to predict subsequent observations, given their own context. One of the salient features of our analysis is that the problems studied here are easier when the dimension of $x_i$ is large; in other words, prediction becomes easier when more context is provided. The proposed methodology is a variant of principal component regression (PCR). Our rigorous analysis sheds new light on PCR. For instance, we show that classical PCR estimators may be inconsistent in the specified setting, unless they are multiplied by a scalar $c > 1$; that is, unless the classical estimator is expanded. This expansion phenomenon appears to be somewhat novel and contrasts with shrinkage methods ($c < 1$), which are far more common in big data analyses.

## 1 Introduction

The phrase "one-shot learning" has been used to describe our ability – as humans – to correctly recognize and understand objects (e.g. images, words) based on very few training examples [1, 2]. Successful one-shot learning requires the learner to incorporate strong contextual information into the learning algorithm (e.g. information on object categories for image classification [1] or "function words" used in conjunction with a novel word and referent in word-learning [3]). Variants of one-shot learning have been widely studied in literature on cognitive science [4, 5], language acquisition (where a great deal of relevant work has been conducted on "fast-mapping") [3, 6–8], and computer vision [1, 9]. Many recent statistical approaches to one-shot learning, which have been shown to perform effectively in a variety of examples, rely on hierarchical Bayesian models, e.g. [1–5, 8].

In this article, we propose a simple latent factor model for one-shot learning with continuous outcomes. We propose effective methods for one-shot learning in this setting, and derive risk approximations that are informative in an asymptotic regime where the number of training examples $n$ is fixed (e.g. $n = 2$) and the number of contextual features for each example $d$ diverges. These approximations provide insight into the significance of various parameters that are relevant for one-shot learning. One important feature of the proposed one-shot setting is that prediction becomes "easier" when $d$ is large – in other words, prediction becomes easier when more context is provided. Binary classification problems that are "easier" when $d$ is large have been previously studied in the literature, e.g. [10–12]; this article may contain the first analysis of this kind with continuous outcomes.

The methods considered in this paper are variants of principal component regression (PCR) [13]. Principal component analysis (PCA) is the cornerstone of PCR. High-dimensional PCA (i.e. large $d$) has been studied extensively in recent literature, e.g. [14–22]. Existing work that is especially relevant for this paper includes that of Lee et al. [19], who studied principal component scores in high dimensions, and work by Hall, Jung, Marron and co-authors [10, 11, 18, 21], who have studied "high dimension, low sample size" data, with fixed $n$ and $d \rightarrow \infty$, in a variety of contexts, including

PCA. While many of these results address issues that are clearly relevant for PCR (e.g. consistency or inconsistency of sample eigenvalues and eigenvectors in high dimensions), their precise implications for high-dimensional PCR are unclear.

In addition to addressing questions about one-shot learning, which motivate the present analysis, the results in this paper provide new insights into PCR in high dimensions. We show that the classical PCR estimator is generally inconsistent in the one-shot learning regime, where $n$ is fixed and $d \to \infty$. To remedy this, we propose a bias-corrected PCR estimator, which is obtained by *expanding* the classical PCR estimator (i.e. multiplying it by a scalar $c > 1$). Risk approximations obtained in Section 5 imply that the bias-corrected estimator is consistent when $n$ is fixed and $d \to \infty$. These results are supported by a simulation study described in Section 7, where we also consider an "oracle" PCR estimator for comparative purposes. It is noteworthy that the bias-corrected estimator is an expanded version of the classical estimator. Shrinkage, which would correspond to multiplying the classical estimator by a scalar $0 \le c < 1$, is a far more common phenomenon in high-dimensional data analysis, e.g. [23–25] (however, expansion is not unprecedented; Lee et al. [19] argued for bias-correction via expansion in the analysis of principal component scores).

## 2   Statistical setting

Suppose that the observed data consists of $(y_1, \mathbf{x}_1), ..., (y_n, \mathbf{x}_n)$, where $y_i \in \mathbb{R}$ is a scalar outcome and $\mathbf{x}_i \in \mathbb{R}^d$ is an associated $d$-dimensional "context" vector, for $i = 1, ..., n$. Suppose that $y_i$ and $\mathbf{x}_i$ are related via

$$
\begin{aligned}
y_i &= h_i \theta + \xi_i \in \mathbb{R}, \ \ h_i \sim N(0, \eta^2), \ \xi_i \sim N(0, \sigma^2), & (1) \\
\mathbf{x}_i &= h_i \gamma \sqrt{d} \mathbf{u} + \boldsymbol{\epsilon}_i \in \mathbb{R}^d, \ \ \boldsymbol{\epsilon}_i \sim N(0, \tau^2 I), \ \ i = 1, ..., n. & (2)
\end{aligned}
$$

The random variables $h_i, \xi_i$ and the random vectors $\boldsymbol{\epsilon}_i = (\epsilon_{i1}, ..., \epsilon_{id})^T$, $1 \le i \le n$, are all assumed to be independent; $h_i$ is a latent factor linking the outcome $y_i$ and the vector $\mathbf{x}_i$; $\xi_i$ and $\boldsymbol{\epsilon}_i$ are random noise. The unit vector $\mathbf{u} = (u_1, ..., u_d)^T \in \mathbb{R}^d$ and real numbers $\theta, \gamma \in \mathbb{R}$ are taken to be non-random. It is implicit in our normalization that the "$\mathbf{x}$-signal" $||h_i \gamma \sqrt{d} \mathbf{u}||^2 \asymp d$ is quite strong.

Observe that $(y_i, \mathbf{x}_i) \sim N(0, V)$ are jointly normal with

$$
V = \left( \begin{array}{cc} \theta^2 \eta^2 + \sigma^2 & \theta \eta^2 \gamma \sqrt{d} \mathbf{u}^T \\ \theta \eta^2 \gamma \sqrt{d} \mathbf{u} & \tau^2 I + \eta^2 \gamma^2 d \mathbf{u} \mathbf{u}^T \end{array} \right). \tag{3}
$$

To further simplify notation in what follows, let $\mathbf{y} = (y_1, ..., y_n)^T = \mathbf{h}\theta + \boldsymbol{\xi} \in \mathbb{R}^n$, where $\mathbf{h} = (h_1, ..., h_n)^T$, $\boldsymbol{\xi} = (\xi_1, ..., \xi_n)^T \in \mathbb{R}^n$, and let $X = (\mathbf{x}_1, ..., \mathbf{x}_n)^T = \gamma \sqrt{d} \mathbf{h} \mathbf{u}^T + \mathsf{E}$, where $\mathsf{E} = (\epsilon_{ij})_{1 \le i \le n, \ 1 \le j \le d}$.

Given the observed data $(\mathbf{y}, X)$, our objective is to devise prediction rules $\hat{y} : \mathbb{R}^d \to \mathbb{R}$ so that the risk

$$
R_V(\hat{y}) = E_V \{ \hat{y}(\mathbf{x}_{new}) - y_{new} \}^2 = E_V \{ \hat{y}(\mathbf{x}_{new}) - h_{new}\theta \}^2 + \sigma^2 \tag{4}
$$

is small, where $(y_{new}, \mathbf{x}_{new}) = (h_{new}\theta + \xi_{new}, h_{new}\gamma \sqrt{d} \mathbf{u} + \boldsymbol{\epsilon}_{new})$ has the same distribution as $(y_i, \mathbf{x}_i)$ and is independent of $(\mathbf{y}, X)$. The subscript "$V$" in $R_V$ and $E_V$ indicates that the parameters $\theta, \eta, \sigma, \tau, \gamma, \mathbf{u}$ are specified by $V$, as in (3); similarly, we will write $P_V(\cdot)$ to denote probabilities with the parameters specified by $V$.

We are primarily interested in identifying methods $\hat{y}$ that perform well (i.e. $R_V(\hat{y})$ is small) in an asymptotic regime whose key features are (i) $n$ is fixed, (ii) $d \to \infty$, (iii) $\sigma^2 \to 0$, and (iv) $\inf \eta^2 \gamma^2 / \tau^2 > 0$. We suggest that this regime reflects a one-shot learning setting, where $n$ is small and $d$ is large (captured by (i)-(ii) from the previous sentence), and there is abundant contextual information for predicting future outcomes (which is ensured by (iii)-(iv)). In a specified asymptotic regime (not necessarily the one-shot regime), we say that a prediction method $\hat{y}$ is *consistent* if $R_V(\hat{y}) \to 0$. Weak consistency is another type of consistency that is considered below. We say that $\hat{y}$ is *weakly consistent* if $|\hat{y} - y_{new}| \to 0$ in probability. Clearly, if $\hat{y}$ is consistent, then it is also weakly consistent.

# 3    Principal component regression

By assumption, the data $(y_i, \mathbf{x}_i)$ are multivariate normal. Thus, $E_V(y_i|\mathbf{x}_i) = \mathbf{x}_i^T \boldsymbol{\beta}$, where $\boldsymbol{\beta} = \theta\gamma\eta^2\sqrt{d}\mathbf{u}/(\tau^2 + \eta^2\gamma^2 d)$. This suggests studying linear prediction rules of the form $\hat{y}(\mathbf{x}_{new}) = \mathbf{x}_{new}^T \hat{\boldsymbol{\beta}}$, for some estimator $\hat{\boldsymbol{\beta}}$ of $\boldsymbol{\beta}$. In this paper, we restrict our attention to linear prediction rules, focusing on estimators related to principal component regression (PCR).

Let $l_1 \geq \cdots \geq l_{n\wedge d} \geq 0$ denote the ordered $n$ largest eigenvalues of $X^T X$ and let $\hat{\mathbf{u}}_1, ..., \hat{\mathbf{u}}_{n\wedge d}$ denote corresponding eigenvectors with unit length; $\hat{\mathbf{u}}_1, ..., \hat{\mathbf{u}}_{n\wedge d}$ are also referred to as the "principal components" of $X$. Let $U_k = (\hat{\mathbf{u}}_1 \; \cdots \; \hat{\mathbf{u}}_k)$ be the $d \times k$ matrix with columns given by $\hat{\mathbf{u}}_1, ..., \hat{\mathbf{u}}_k$, for $1 \leq k \leq n \wedge d$. In its most basic form, principal component regression involves regressing $\mathbf{y}$ on $XU_k$ for some (typically small) $k$, and taking $\hat{\boldsymbol{\beta}} = U_k(U_k^T X^T X U_k)^{-1} U_k^T X^T \mathbf{y}$. In the problem considered here the predictor covariance matrix $\text{Cov}(\mathbf{x}_i) = \tau^2 I + \eta^2\gamma^2 d\mathbf{u}\mathbf{u}^T$ has a single eigenvalue larger than $\tau^2$ and the corresponding eigenvector is parallel to $\boldsymbol{\beta}$. Thus, it is natural to restrict our attention to PCR with $k = 1$; more explicitly, consider

$$\hat{\boldsymbol{\beta}}_{pcr} = \frac{\hat{\mathbf{u}}_1^T X^T \mathbf{y}}{\hat{\mathbf{u}}_1^T X^T X \hat{\mathbf{u}}_1} \hat{\mathbf{u}}_1 = \frac{1}{l_1} \hat{\mathbf{u}}_1^T X^T \mathbf{y} \hat{\mathbf{u}}_1. \tag{5}$$

In the following sections, we study consistency and risk properties of $\hat{\boldsymbol{\beta}}_{pcr}$ and related estimators.

# 4    Weak consistency and big data with $n = 2$

Before turning our attention to risk approximations for PCR in Section 5 below (which contains the paper's main technical contributions), we discuss weak consistency in the one-shot asymptotic regime, devoting special attention to the case where $n = 2$. This serves at least two purposes. First, it provides an illustrative warm-up for the more complex risk bounds obtained in Section 5. Second, it will become apparent below that the risk of the consistent PCR methods studied in this paper depends on inverse moments of $\chi^2$ random variables. For very small $n$, these inverse moments do not exist and, consequently, the risk of the associated prediction methods may be infinite. The main implication of this is that the risk bounds in Section 5 require $n \geq 9$ to ensure their validity. On the other hand, the weak consistency results obtained in this section are valid for all $n \geq 2$.

## 4.1    Heuristic analysis for $n = 2$

Recall the PCR estimator (5) and let $\hat{y}_{pcr}(\mathbf{x}) = \mathbf{x}^T \hat{\boldsymbol{\beta}}_{pcr}$ be the associated linear prediction rule. For $n = 2$, the largest eigenvalue of $X^T X$ and the corresponding eigenvector are given by simple explicit formulas:

$$l_1 = \frac{1}{2}\left\{||\mathbf{x}_1||^2 + ||\mathbf{x}_2||^2 + \sqrt{(||\mathbf{x}_1||^2 - ||\mathbf{x}_2||^2)^2 + 4(\mathbf{x}_1^T\mathbf{x}_2)^2}\right\}$$

and $\hat{\mathbf{u}}_1 = \hat{\mathbf{v}}_1/||\hat{\mathbf{v}}_1||^2$, where

$$\hat{\mathbf{v}}_1 = \frac{1}{2\mathbf{x}_1^T\mathbf{x}_2}\left\{||\mathbf{x}_1||^2 - ||\mathbf{x}_2||^2 + \sqrt{(||\mathbf{x}_1||^2 - ||\mathbf{x}_2||^2)^2 + 4(\mathbf{x}_1^T\mathbf{x}_2)^2}\right\}\mathbf{x}_1 + \mathbf{x}_2.$$

These expressions for $l_1$ and $\hat{\mathbf{u}}_1$ yield an explicit expression for $\hat{\boldsymbol{\beta}}_{pcr}$ when $n = 2$ and facilitate a simple heuristic analysis of PCR, which we undertake in this subsection. This analysis suggests that $\hat{y}_{pcr}$ is *not* consistent when $\sigma^2 \to 0$ and $d \to \infty$ (at least for $n = 2$). However, the analysis also suggests that consistency can be achieved by multiplying $\hat{\boldsymbol{\beta}}_{pcr}$ by a scalar $c \geq 1$; that is, by *expanding* $\hat{\boldsymbol{\beta}}_{pcr}$. This observation leads us to consider and rigorously analyze a bias-corrected PCR method, which we ultimately show is consistent in fixed $n$ settings, if $\sigma^2 \to 0$ and $d \to \infty$. On the other hand, it will also be shown below that $\hat{y}_{pcr}$ is inconsistent in one-shot asymptotic regimes.

For large $d$, the basic approximations $||\mathbf{x}_i||^2 \approx \gamma^2 dh_1^2 + \tau^2 d$ and $\mathbf{x}_1^T\mathbf{x}_2 \approx \gamma^2 dh_ih_j$ lead to the following approximation for $\hat{y}_{pcr}(\mathbf{x}_{new})$:

$$\hat{y}_{pcr}(\mathbf{x}_{new}) = \mathbf{x}_{new}^T \hat{\boldsymbol{\beta}}_{pcr} \approx \frac{\gamma^2(h_1^2 + h_2^2)}{\gamma^2(h_1^2 + h_2^2) + \tau^2} h_{new}\theta + e_{pcr}, \tag{6}$$

where
$$e_{pcr} = \frac{\gamma^2 h_{new}}{\{\gamma^2 d(h_1^2 + h_2^2) + \tau^2 d\}^2} \hat{\mathbf{u}}_1^T X^T \boldsymbol{\xi}.$$
Thus,
$$\hat{y}_{pcr}(\mathbf{x}_{new}) - y_{new} \approx -\frac{\tau^2}{\gamma^2(h_1^2 + h_2^2) + \tau^2} h_{new}\theta + e_{pcr} - \xi_{new}. \tag{7}$$

The second and third terms on the right-hand side in (7), $e_{pcr} - \xi_{new}$, represent a random error that vanishes as $d \to \infty$ and $\sigma^2 \to 0$. On the other hand, the first term on the right-hand side in (7), $-\tau^2 h_{new}\theta/\{\gamma^2(h_1^2 + h_2^2) + \tau^2\}$, is a bias term that is, in general, non-zero when $d \to \infty$ and $\sigma^2 \to 0$; in other words $\hat{y}_{pcr}$ is inconsistent. This bias is apparent in the expression for $\hat{y}_{pcr}(\mathbf{x}_{new})$ given in (6); in particular, the first term on the right-hand side of (6) is typically smaller than $h_{new}\theta$. One way to correct for the bias of $\hat{y}_{pcr}$ is to multiply $\hat{\boldsymbol{\beta}}_{pcr}$ by
$$\frac{l_1}{l_1 - l_2} \approx \frac{\gamma^2(h_1^2 + h_2^2) + \tau^2}{\gamma^2(h_1^2 + h_2^2)} \geq 1,$$
where
$$l_2 = \frac{1}{2}\left\{ ||\mathbf{x}_1||^2 + ||\mathbf{x}_2||^2 - \sqrt{(||\mathbf{x}_1||^2 - ||\mathbf{x}_2||^2)^2 + 4(\mathbf{x}_1^T\mathbf{x}_2)^2} \right\} \approx \tau^2 d$$
is the second-largest eigenvalue of $X^T X$. Define the bias-corrected principal component regression estimator
$$\hat{\boldsymbol{\beta}}_{bc} = \frac{l_1}{l_1 - l_2}\hat{\boldsymbol{\beta}}_{pcr} = \frac{1}{l_1 - l_2}\hat{\mathbf{u}}_1^T X^T \mathbf{y}$$
and let $\hat{y}_{bc}(\mathbf{x}) = \mathbf{x}^T \hat{\boldsymbol{\beta}}_{bc}$ be the associated linear prediction rule. Then $\hat{y}_{bc}(\mathbf{x}_{new}) = \mathbf{x}_{new}^T \hat{\boldsymbol{\beta}}_{bc} \approx h_{new}\theta + e_{bc}$, where
$$e_{bc} = \frac{h_{new}}{\{\gamma^2(h_1^2 + h_2^2) + \tau^2\}(h_1^2 + h_2^2)d^2}\hat{\mathbf{u}}_1^T X^T \boldsymbol{\xi}.$$
One can check that if $d \to \infty$, $\sigma^2 \to 0$ and $\theta, \eta^2, \gamma^2, \tau^2$ are well-behaved (e.g. contained in a compact subset of $(0, \infty)$), then $\hat{y}_{bc}(\mathbf{x}_{new}) - y_{new} \approx e_{bc} \to 0$ in probability; in other words, $\hat{y}_{bc}$ is weakly consistent. Indeed, weak consistency of $\hat{y}_{bc}$ follows from Theorem 1 below. On the other hand, note that $E|e_{bc}| = \infty$. This suggests that $R_V(\hat{y}_{bc}) = \infty$, which in fact may be confirmed by direct calculation. Thus, when $n = 2$, $\hat{y}_{bc}$ is weakly consistent, but not consistent.

## 4.2 Weak consistency for bias-corrected PCR

Now suppose that $n \geq 2$ is arbitrary and that $d \geq n$. Define the bias-corrected PCR estimator
$$\hat{\boldsymbol{\beta}}_{bc} = \frac{l_1}{l_1 - l_n}\hat{\boldsymbol{\beta}}_{pcr} = \frac{1}{l_1 - l_n}\hat{\mathbf{u}}_1^T X^T \mathbf{y}\hat{\mathbf{u}}_1 \tag{8}$$
and the associated linear prediction rule $\hat{y}_{bc}(\mathbf{x}) = \mathbf{x}^T \hat{\boldsymbol{\beta}}_{bc}$. The main weak consistency result of the paper is given below.

**Theorem 1.** *Suppose that $n \geq 2$ is fixed and let $C \subseteq (0, \infty)$ be a compact set. Let $r > 0$ be an arbitrary but fixed positive real number. Then*
$$\lim_{\substack{d \to \infty \\ \sigma^2 \to 0}} \sup_{\substack{\theta, \eta, \tau, \gamma \in C \\ \mathbf{u} \in \mathbb{R}^d}} P_V\left\{|\hat{y}_{bc}(\mathbf{x}_{new}) - y_{new}| > r\right\} = 0. \tag{9}$$

*On the other hand,*
$$\liminf_{\substack{d \to \infty \\ \sigma^2 \to 0}} \inf_{\substack{\theta, \eta, \tau, \gamma \in C \\ \mathbf{u} \in \mathbb{R}^d}} P_V\left\{|\hat{y}_{pcr}(\mathbf{x}_{new}) - y_{new}| > r\right\} > 0. \tag{10}$$

A proof of Theorem 1 follows easily upon inspection of the proof of Theorem 2, which may be found in the Supplementary Material. Theorem 1 implies that in the specified fixed $n$ asymptotic setting, bias-corrected PCR is weakly consistent (9) and that the more standard PCR method $\hat{y}_{pcr}$ is inconsistent (10). Note that the condition $\theta, \eta, \tau, \gamma \in C$ in (9) ensures that the x-data signal-to-noise ratio $\eta^2\gamma^2/\tau^2$ is bounded away from 0. In (8), it is noteworthy that $l_1/(l_1 - l_n) \geq 1$: in order to achieve (weak) consistency, the bias corrected estimator $\hat{\boldsymbol{\beta}}_{bc}$ is obtained by *expanding* $\hat{\boldsymbol{\beta}}_{pcr}$. By contrast, *shrinkage* is a far more common method for obtaining improved estimators in many regression and prediction settings (the literature on shrinkage estimation is vast, perhaps beginning with [23]).

# 5 Risk approximations and consistency

In this section, we present risk approximations for $\hat{y}_{pcr}$ and $\hat{y}_{bc}$ that are valid when $n \geq 9$. A more careful analysis may yield approximations that are valid for smaller $n$; however, this is not pursued further here.

**Theorem 2.** *Let $W_n \sim \chi_n^2$ be a chi-squared random variable with $n$ degrees of freedom.*

(a) *If $n \geq 9$ and $d \geq 1$, then*

$$
R_V(\hat{y}_{pcr}) = \sigma^2 \left[ 1 + E \left\{ \frac{\eta^4 \gamma^4 W_n}{(\eta^2 \gamma^2 W_n + \tau^2)^2} \right\} \right] + O \left( \frac{\sigma^2}{n} \sqrt{\frac{n}{d+n}} \right)
$$
$$
+ \theta^2 \eta^2 E_V \left\{ (\mathbf{u}^T \hat{\mathbf{u}}_1)^2 - 1 \right\}^2 + O \left( \frac{\theta^2 \eta^2 \tau^2}{\eta^2 \gamma^2 d + \tau^2} \right).
$$

(b) *If $d \geq n \geq 9$, then*

$$
R_V(\hat{y}_{bc}) = \sigma^2 \left\{ 1 + E \left( \frac{\eta^2 \gamma^2}{\eta^2 \gamma^2 W_n + \tau^2 \sqrt{n/d}} \right) \right\} + O \left\{ \frac{\sigma^2}{\sqrt{dn}} \left( \frac{\tau^2}{\eta^2 \gamma^2 n + \tau^2 \sqrt{n/d}} \right) \right\}
$$
$$
+ \theta^2 \eta^2 E_V \left\{ \frac{l_1}{l_1 - l_n} (\mathbf{u}^T \hat{\mathbf{u}}_1)^2 - 1 \right\}^2 \tag{11}
$$
$$
+ \frac{\theta^2 \eta^2 \tau^2}{\eta^2 \gamma^2 d + \tau^2} \left\{ 1 + E \left( \frac{\eta^2 \gamma^2 + \tau^2}{\eta^2 \gamma^2 W_n + \tau^2 \sqrt{n/d}} \right) \right\} \tag{12}
$$
$$
+ O \left[ \frac{\theta^2 \eta^2 \tau^2}{\eta^2 \gamma^2 d + \tau^2} \sqrt{\frac{n}{d}} \left\{ \frac{\tau^2}{\eta^2 \gamma^2 n + \tau^2 \sqrt{n/d}} + \frac{\tau^4}{(\eta^2 \gamma^2 n + \tau^2 \sqrt{n/d})^2} \right\} \right].
$$

A proof of Theorem 2 (along with intermediate lemmas and propositions) may be found in the Supplementary Material. The necessity of the more complex error term in Theorem 2 (b) (as opposed to that in part (a)) will become apparent below.

When $d$ is large, $\sigma^2$ is small, and $\theta, \eta, \tau, \gamma \in C$, for some compact subset $C \subseteq (0, \infty)$, Theorem 2 suggests that

$$
R_V(\hat{y}_{pcr}) \approx \theta^2 \eta^2 E_V \left\{ (\mathbf{u}^T \hat{\mathbf{u}}_1)^2 - 1 \right\}^2,
$$
$$
R_V(\hat{y}_{bc}) \approx \theta^2 \eta^2 E_V \left\{ \frac{l_1}{l_1 - l_n} (\mathbf{u}^T \hat{\mathbf{u}}_1)^2 - 1 \right\}^2.
$$

Thus, consistency of $\hat{y}_{pcr}$ and $\hat{y}_{bc}$ in the one-shot regime hinges on asymptotic properties of $E_V \{ (\mathbf{u}^T \hat{\mathbf{u}}_1)^2 - 1 \}^2$ and $E_V \{ l_1 / (l_1 - l_n)(\mathbf{u}^T \hat{\mathbf{u}}_1)^2 - 1 \}^2$. The following proposition is proved in the Supplementary Material.

**Proposition 1.** *Let $W_n \sim \chi_n^2$ be a chi-squared random variable with $n$ degrees of freedom.*

(a) *If $n \geq 9$ and $d \geq 1$, then*

$$
E_V \left\{ (\mathbf{u}^T \hat{\mathbf{u}}_1)^2 - 1 \right\}^2 = E \left( \frac{\tau^2}{\eta^2 \gamma^2 W_n + \tau^2} \right)^2 + O \left( \sqrt{\frac{n}{d+n}} \right).
$$

(b) *If $d \geq n \geq 9$, then*

$$
E_V \left\{ \frac{l_1}{l_1 - l_n} (\mathbf{u}^T \hat{\mathbf{u}}_1)^2 - 1 \right\}^2 = O \left\{ \frac{n}{d} \cdot \frac{\tau^4}{(\eta^2 \gamma^2 n + \tau^2 \sqrt{n/d})^2} \right\}.
$$

Proposition 1 (a) implies that in the one-shot regime, $E_V \{ (\mathbf{u}^T \hat{\mathbf{u}}_1)^2 - 1 \}^2 \to E \{ \tau^2 / (\eta^2 \gamma^2 W_n + \tau^2)^2 \} \neq 0$; by Theorem 2 (a), it follows that $\hat{y}_{pcr}$ is inconsistent. On the other hand, Proposition 1 (b) implies that $E_V \left\{ l_1 / (l_1 - l_n)(\mathbf{u}^T \hat{\mathbf{u}}_1)^2 - 1 \right\}^2 \to 0$ in the one-shot regime; thus, by Theorem 2 (b), $\hat{y}_{bc}$ is consistent. These results are summarized in Corollary 1, which follows immediately from Theorem 2 and Proposition 1.

**Corollary 1.** *Suppose that $n \geq 9$ is fixed and let $C \subseteq (0, \infty)$ be a compact set. Let $W_n \sim \chi_n^2$ be a chi-squared random variable with $n$ degrees of freedom. Then*

$$\lim_{\substack{d \to \infty \\ \sigma^2 \to 0}} \sup_{\substack{\theta, \eta, \tau, \gamma \in C \\ \mathbf{u} \in \mathbb{R}^d}} \left| R_V(\hat{y}_{pcr}) - \theta^2 \eta^2 E \left( \frac{\tau^2}{\eta^2 \gamma^2 W_n + \tau^2} \right)^2 \right| = 0.$$

*and*

$$\lim_{\substack{d \to \infty \\ \sigma^2 \to 0}} \sup_{\substack{\theta, \eta, \tau, \gamma \in C \\ \mathbf{u} \in \mathbb{R}^d}} R_V(\hat{y}_{bc}) = 0.$$

For fixed $n$, and $\inf \eta^2 \gamma^2 / \tau^2 > 0$, the bound in Proposition 1 (b) is of order $1/d$. This suggests that both terms (11)-(12) in Theorem 2 (b) have similar magnitude and, consequently, are both necessary to obtain accurate approximations for $R_V(\hat{y}_{bc})$. (It may be desirable to obtain more accurate approximations for $E_V \left\{ l_1/(l_1 - l_n)(\mathbf{u}^T \hat{\mathbf{u}}_1)^2 - 1 \right\}^2$; this could potentially be leveraged to obtain better approximations for $R_V(\hat{y}_{bc})$.) In Theorem 2 (a), the only non-vanishing term in the one-shot approximation for $R_V(\hat{y}_{pcr})$ involves $E_V \{ (\mathbf{u}^T \hat{\mathbf{u}}_1)^2 - 1 \}^2$; this helps to explain the relative simplicity of this approximation, in comparison with Theorem 2 (b).

Theorem 2 and Proposition 1 give risk approximations that are valid for all $d$ and $n \geq 9$. However, as illustrated by Corollary 1, these approximations are most effective in a one-shot asymptotic setting, where $n$ is fixed and $d$ is large. In the one-shot regime, standard concepts, such as sample complexity – roughly, the sample size $n$ required to ensure a certain risk bound – may be of secondary importance. Alternatively, in a one-shot setting, one might be more interested in metrics like "feature complexity": the number of features $d$ required to ensure a given risk bound. Approximate feature complexity for $\hat{y}_{bc}$ is easily computed using Theorem 2 and Proposition 1 (clearly, feature complexity depends heavily on model parameters, such as $\theta$, the $y$-data noise level $\sigma^2$, and the $\mathbf{x}$-data signal-to-noise ratio $\eta^2 \gamma^2 / \tau^2$).

## 6    An oracle estimator

In this section, we discuss a third method related to $\hat{y}_{pcr}$ and $\hat{y}_{bc}$, which relies on information that is typically not available in practice. Thus, this method is usually non-implementable; however, we believe it is useful for comparative purposes.

Recall that both $\hat{y}_{bc}$ and $\hat{y}_{pcr}$ depend on the first principal component $\hat{\mathbf{u}}_1$, which may be viewed as an estimate of $\mathbf{u}$. If an oracle provides knowledge of $\mathbf{u}$ in advance, then it is natural to consider the oracle PCR estimator

$$\hat{\boldsymbol{\beta}}_{or} = \frac{\mathbf{u}^T X^T \mathbf{y}}{\mathbf{u}^T X^T X \mathbf{u}} \mathbf{u}$$

and the associated linear prediction rule $\hat{y}_{or}(\mathbf{x}) = \mathbf{x}^T \hat{\boldsymbol{\beta}}_{or}$. A basic calculation yields the following result.

**Proposition 2.** *If $n \geq 3$, then*

$$R_V(\hat{y}_{or}) = \left( \sigma^2 + \frac{\theta^2 \eta^2 \tau^2}{\eta^2 \gamma^2 d + \tau^2} \right) \left( 1 + \frac{1}{n-2} \right).$$

Clearly, $\hat{y}_{or}$ is consistent in the one-shot regime: if $C \subseteq (0, \infty)$ is compact and $n \geq 3$ is fixed, then

$$\lim_{\substack{d \to \infty \\ \sigma^2 \to 0}} \sup_{\substack{\theta, \eta, \tau, \gamma \in C \\ \mathbf{u} \in \mathbb{R}^d}} R_V(\hat{y}_{or}) = 0.$$

## 7    Numerical results

In this section, we describe the results of a simulation study where we compared the performance of $\hat{y}_{pcr}$, $\hat{y}_{bc}$, and $\hat{y}_{or}$. We fixed $\theta = 4$, $\sigma^2 = 1/10$, $\eta^2 = 4$, $\gamma^2 = 1/4$, $\tau^2 = 1$, and $\mathbf{u} = (1, 0, ...., 0) \in$

$\mathbb{R}^d$ and simulated 1000 independent datasets with various $d, n$. Observe that $\eta^2\gamma^2/\tau^2 = 1$. For each simulated dataset, we computed $\hat{\boldsymbol{\beta}}_{pcr}$, $\hat{\boldsymbol{\beta}}_{bc}$, $\hat{\boldsymbol{\beta}}_{or}$ and the corresponding conditional prediction error

$$
\begin{aligned}
R_V(\hat{y}|\mathbf{y}, X) &= E\left[\{\hat{y}(\mathbf{x}_{new}) - y_{new}\}^2 \big| \mathbf{y}, X\right] \\
&= (\hat{\boldsymbol{\beta}} - \boldsymbol{\beta})^T(\tau^2 I + \eta^2\gamma^2 d\mathbf{u}\mathbf{u}^T)(\hat{\boldsymbol{\beta}} - \boldsymbol{\beta}) + \sigma^2 + \frac{\theta^2\eta^2}{\psi^2 d + 1},
\end{aligned}
$$

for $\hat{y} = \hat{y}_{pcr}$, $\hat{y}_{bc}$, $\hat{y}_{or}$. The empirical prediction error for each method $\hat{y}$ was then computed by averaging $R_V(\hat{y}|\mathbf{y}, X)$ over all 1000 simulated datasets. We also computed the "theoretical" prediction error for each method, using the results from Sections 5-6, where appropriate. More specifically, for $\hat{y}_{pcr}$ and $\hat{y}_{bc}$, we used the leading terms of the approximations in Theorem 2 and Proposition 1 to obtain the theoretical prediction error; for $\hat{y}_{or}$, we used the formula given in Proposition 2 (see Table 1 for more details). Finally, we computed the relative error between the empirical prediction error

Table 1: Formulas for theoretical prediction error used in simulations (derived from Theorem 2 and Propositions 1-2). Expectations in theoretical prediction error expressions for $\hat{y}_{pcr}$ and $\hat{y}_{bc}$ were computed empirically.

| | Theoretical prediction error formula |
|---|---|
| $\hat{y}_{pcr}$ | $\sigma^2\left[1 + E\left\{\frac{\eta^4\gamma^4 W_n}{(\eta^2\gamma^2 W_n + \tau^2)^2}\right\}\right] + \theta^2\eta^2 E\left(\frac{\tau^2}{\eta^2\gamma^2 W_n + \tau^2}\right)^2$ |
| $\hat{y}_{bc}$ | $\sigma^2\left\{1 + E\left(\frac{\eta^2\gamma^2}{\eta^2\gamma^2 W_n + \tau^2\sqrt{n/d}}\right)\right\} + \theta^2\eta^2 E_V\left\{\frac{l_1}{l_1 - l_n}(\mathbf{u}^T\hat{\mathbf{u}}_1)^2 - 1\right\}^2$ $+ \frac{\theta^2\eta^2\tau^2}{\eta^2\gamma^2 d + \tau^2}\left\{1 + E\left(\frac{\eta^2\gamma^2 + \tau^2}{\eta^2\gamma^2 W_n + \tau^2\sqrt{n/d}}\right)\right\}$ |
| $\hat{y}_{or}$ | $\left(\sigma^2 + \frac{\theta^2\eta^2\tau^2}{\eta^2\gamma^2 d + \tau^2}\right)\left(1 + \frac{1}{n-2}\right)$ |

and the theoretical prediction error for each method,

$$
\text{Relative Error} = \left|\frac{(\text{Empirical PE}) - (\text{Theoretical PE})}{\text{Empirical PE}}\right| \times 100\%.
$$

Table 2: $d = 500$. Prediction error for $\hat{y}_{pcr}$ (PCR), $\hat{y}_{bc}$ (Bias-corrected PCR), and $\hat{y}_{or}$ (oracle). Relative error for comparing Empirical PE and Theoretical PE is given in parentheses. "NA" indicates that Theoretical PE values are unknown.

| | | PCR | Bias-corrected PCR | Oracle |
|---|---|---|---|---|
| $n = 2$ | Empirical PE | 18.7967 | 4.8668 | 1.5836 |
| | Theoretical PE (Relative Error) | NA | $\infty$ ($\infty$) | $\infty$ ($\infty$) |
| $n = 4$ | Empirical PE | 6.4639 | 0.8023 | 0.3268 |
| | Theoretical PE (Relative Error) | NA | NA | 0.3416 (4.53%) |
| $n = 9$ | Empirical PE | 1.4187 | 0.3565 | 0.2587 |
| | Theoretical PE (Relative Error) | 1.2514 (11.79%) | 0.2857 (19.86%) | 0.2603 (0.62%) |
| $n = 20$ | Empirical PE | 0.4513 | 0.2732 | 0.2398 |
| | Theoretical PE (Relative Error) | 0.2987 (33.81%) | 0.2497 (8.60%) | 0.2404 (0.25%) |

The results of the simulation study are summarized in Tables 2-3. Observe that $\hat{y}_{bc}$ has smaller empirical prediction error than $\hat{y}_{pcr}$ in every setting considered in Tables 2-3, and $\hat{y}_{bc}$ substantially outperforms $\hat{y}_{pcr}$ in most settings. Indeed, the empirical prediction error for $\hat{y}_{bc}$ when $n = 9$ is smaller than that of $\hat{y}_{pcr}$ when $n = 20$ (for both $d = 500$ and $d = 5000$); in other words, $\hat{y}_{bc}$ outperforms $\hat{y}_{pcr}$, even when $\hat{y}_{pcr}$ has more than twice as much training data. Additionally, the empirical prediction error of $\hat{y}_{bc}$ is quite close to that of the oracle method $\hat{y}_{or}$, especially when $n$ is relatively large. These results highlight the effectiveness of the bias-corrected PCR method $\hat{y}_{bc}$ in settings where $\sigma^2$ and $n$ are small, $\eta^2\gamma^2/\tau^2$ is substantially larger than 0, and $d$ is large.

For $n = 2, 4$, theoretical prediction error is unavailable in some instances. Indeed, while Proposition 2 and the discussion in Section 4 imply that if $n = 2$, then $R_V(\hat{y}_{bc}) = R_V(\hat{y}_{or}) = \infty$, we have not

Table 3: $d = 5000$. Prediction error for $\hat{y}_{pcr}$ (PCR), $\hat{y}_{bc}$ (Bias-corrected PCR), and $\hat{y}_{or}$ (oracle). Relative error comparing Empirical PE and Theoretical PE is given in parentheses. "NA" indicates that Theoretical PE values are unknown.

| | | PCR | Bias-corrected PCR | Oracle |
|---|---|---|---|---|
| $n = 2$ | Empirical PE | 17.9564 | 2.0192 | 1.0316 |
| | Theoretical PE  (Relative Error) | NA | $\infty$  ($\infty$) | $\infty$  ($\infty$) |
| $n = 4$ | Empirical PE | 6.1220 | 0.2039 | 0.1637 |
| | Theoretical PE  (Relative Error) | NA | NA | 0.1692 (3.36%) |
| $n = 9$ | Empirical PE | 1.2274 | 0.1378 | 0.1281 |
| | Theoretical PE  (Relative Error) | 1.2485  (1.72%) | 0.1314 (4.64%) | 0.1289 (0.62%) |
| $n = 20$ | Empirical PE | 0.3150 | 0.1226 | 0.1189 |
| | Theoretical PE  (Relative Error) | 0.2997  (4.86%) | 0.1200 (2.12%) | 0.1191 (0.17%) |

pursued an expression for $R_V(\hat{y}_{pcr})$ when $n = 2$ (it appears that $R_V(\hat{y}_{pcr}) < \infty$); furthermore, the approximations in Theorem 2 for $R_V(\hat{y}_{pcr})$, $R_V(\hat{y}_{bc})$ do not apply when $n = 4$. In instances where theoretical prediction error is available, is finite, and $d = 500$, the relative error between empirical and theoretical prediction error for $\hat{y}_{pcr}$ and $\hat{y}_{bc}$ ranges from 8.60%-33.81%; for $d = 5000$, it ranges from 1.72%-4.86%. Thus, the accuracy of the theoretical prediction error formulas tends to improve as $d$ increases, as one would expect. Further improved measures of theoretical prediction error for $\hat{y}_{pcr}$ and $\hat{y}_{bc}$ could potentially be obtained by refining the approximations in Theorem 2 and Proposition 1.

## 8   Discussion

In this article, we have proposed bias-corrected PCR for consistent one-shot learning in a simple latent factor model with continuous outcomes. Our analysis was motivated by problems in one-shot learning, as discussed in Section 1. However, the results in this paper may also be relevant for other applications and techniques related to high-dimensional data analysis, such as those involving reproducing kernel Hilbert spaces. Furthermore, our analysis sheds new light on PCR, a long-studied method for regression and prediction.

Many open questions remain. For instance, consider the semi-supervised setting, where additional unlabeled data $\mathbf{x}_{n+1}, ..., \mathbf{x}_N$ is available, but the corresponding $y_i$'s are not provided. Then the additional $\mathbf{x}$-data could be used to obtain a better estimate of the first principal component $\mathbf{u}$ and perhaps devise a method whose performance is closer to that of the oracle procedure $\hat{y}_{or}$ (indeed, $\hat{y}_{or}$ may viewed as a semi-supervised procedure that utilizes an infinite amount of unlabeled data to exactly identify $\mathbf{u}$). Is bias-correction via inflation necessary in this setting? Presumably, bias-correction is not needed if $N$ is large enough, but can this be made more precise? The simulations described in the previous section indicate that $\hat{y}_{bc}$ outperforms the uncorrected PCR method $\hat{y}_{pcr}$ in settings where twice as much *labeled* data is available for $\hat{y}_{pcr}$. This suggests that role of bias-correction will remain significant in the semi-supervised setting, where additional unlabeled data (which is less informative than labeled data) is available. Related questions involving transductive learning [26, 27] may also be of interest for future research.

A potentially interesting extension of the present work involves multi-factor models. As opposed to the single-factor model (1)-(2), one could consider a more general $k$-factor model, where $y_i = \mathbf{h}_i^T \boldsymbol{\theta} + \xi_i$ and $\mathbf{x}_i = S\mathbf{h}_i + \boldsymbol{\epsilon}_i$; here $\mathbf{h}_i = (h_{i1}, ..., h_{ik})^T \in \mathbb{R}^k$ is a multivariate normal random vector (a $k$-dimensional factor linking $y_i$ and $\mathbf{x}_i$), $\boldsymbol{\theta} = (\theta_1, ..., \theta_k)^T \in \mathbb{R}^k$, and $S = \sqrt{d}(\gamma_1 \mathbf{u}_1 \cdots \gamma_k \mathbf{u}_k)$ is a $k \times d$ matrix, with $\gamma_1, ..., \gamma_k \in \mathbb{R}$ and unit vectors $\mathbf{u}_1, ..., \mathbf{u}_k \in \mathbb{R}^d$. It may also be of interest to work on relaxing the distributional (normality) assumptions made in this paper. Finally, we point out that the results in this paper could potentially be used to develop flexible probit (latent variable) models for one-shot classification problems.

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
