[Supplementary Material]

# Supplementary material for "One-shot learning and big data with $n = 2$"

**Lee H. Dicker**
Rutgers University
Piscataway, NJ
ldicker@stat.rutgers.edu

**Dean P. Foster**
University of Pennsylvania
Philadelphia, PA
dean@foster.net

In this supplementary document, we retain all notation from the main text. Furthermore, before proceeding, we introduce some additional concepts and notation that will be useful below. Observe that there is an $n \times d$ matrix $Z = (z_{ij})$ with iid $N(0,1)$ entries satisfying

$$XX^T = Z(\tau^2 I + \eta^2\gamma^2 d\mathbf{u}\mathbf{u}^T)Z^T = \tau^2 ZZ^T + \eta^2\gamma^2 d\mathbf{z}_1\mathbf{z}_1^T.$$

where $\mathbf{z}_j$ is the $j$-th column of $Z$. Let $\lambda_1 \geq \cdots \geq \lambda_n \geq 0$ denote the ordered eigenvalues of $ZZ^T$. Finally, let $\psi^2 = \eta^2\gamma^2/\tau^2$ denote the **x**-data signal-to-noise ratio.

## Appendix A: Proof of Theorem 2 (a)

It is straightforward to check that

$$R_V(\hat{y}_{pcr}) = \sum_{j=1}^{9} I_j, \tag{1}$$

where

$$
\begin{aligned}
I_1 &= \sigma^2, \\
I_2 &= \sigma^2 E_V\left(\frac{\tau^2}{l_1}\right), \\
I_3 &= \sigma^2\psi^2 d E_V\left\{\frac{\tau^2}{l_1}(\mathbf{u}^T\hat{\mathbf{u}}_1)^2\right\}, \\
I_4 &= \theta^2\eta^2\left(\frac{\psi^2 d}{\psi^2 d + 1}\right)^2 E_V\left\{(\mathbf{u}^T\hat{\mathbf{u}}_1)^2 - 1\right\}^2, \\
I_5 &= \frac{\theta^2\eta^2}{(\psi^2 d + 1)^2}, \\
I_6 &= -2\theta^2\eta^2\frac{\psi^2 d}{(\psi^2 d + 1)^2}E_V\left\{(\mathbf{u}^T\hat{\mathbf{u}}_1)^2 - 1\right\}, \\
I_7 &= \frac{\theta^2\eta^2}{\psi^2 d + 1}E_V\left(\frac{\tau^2}{l_1}\right), \\
I_8 &= \theta^2\eta^2\frac{\psi^2 d}{\psi^2 d + 1}E\left\{\frac{\tau^2}{l_1}(\mathbf{u}^T\hat{\mathbf{u}}_1)^2\right\}, \\
I_9 &= \theta^2\eta^2\frac{\psi^2 d}{(\psi^2 d + 1)^2}E_V(\mathbf{u}^T\hat{\mathbf{u}}_1)^2.
\end{aligned}
$$

Lemmas A1-A3 and Corollary A1 below give bounds on the expectations appearing in $I_1, ..., I_9$, which lead to the following approximations

$$I_2 = O\left(\frac{\sigma^2}{\psi^2 dn + d + n}\right),$$

$$
\begin{aligned}
I_3 &= \sigma^2 E\left\{\frac{\psi^4\|\mathbf{z}_1\|^2}{(\psi^2\|\mathbf{z}_1\|^2+1)^2}\right\} + O\left(\frac{\sigma^2}{n}\sqrt{\frac{n}{d+n}}\right), \\
I_6 &= O\left\{\theta^2\eta^2\frac{\psi^2 d}{(\psi^2 d+1)^2}\left(\frac{d+n}{\psi^2 dn+d+n}\right)\right\}, \\
I_7 &= O\left\{\frac{\theta^2\eta^2}{\psi^2 d+1}\left(\frac{1}{\psi^2 dn+d+n}\right)\right\}, \\
I_8 &= O\left\{\frac{\theta^2\eta^2}{\psi^2 d+1}\left(\frac{\psi^2 d}{\psi^2 dn+d+n}\right)\right\}, \\
I_9 &= O\left\{\frac{\theta^2\eta^2}{\psi^2 d+1}\left(\frac{\psi^2 d}{\psi^2 d+1}\right)\right\}.
\end{aligned}
$$

The theorem follows by combining these approximations with (1).

The rest of this appendix is devoted to the statement and proof of various lemmas used to obtain the above approximations.

**Lemma A1.** *Let $k > 0$ be a fixed positive number. For $n \geq 2k+1$ and $d \geq 1$,*

$$
E_V\left(\frac{\tau^2}{l_1}\right)^k = O\left\{\left(\frac{1}{\psi^2 dn+d+n}\right)^k\right\} \tag{2}
$$

*If $n \geq 9$ and $d \geq 1$, then*

$$
E_V\left(\frac{\tau^2}{l_1}\right) = E\left\{\frac{1}{(\psi^2 d+1)\|\mathbf{z}_1\|^2+d}\right\} + O\left\{\frac{d+n}{(\psi^2 dn+d+n)^2}\left(\frac{n}{d+n}\right)^{1/2}\right\} \tag{3}
$$

*Proof.* Let $\tilde{\lambda}_n$ denote the smallest eigenvalue of $\sum_{j=2}^{d}\mathbf{z}_j\mathbf{z}_j^T$. Then

$$
(\psi^2 d+1)\|\mathbf{z}_1\|^2 + \tilde{\lambda}_n \leq \frac{l_1}{\tau^2} \leq (\psi^2 d+1)\|\mathbf{z}_1\|^2 + \tilde{\lambda}_1. \tag{4}
$$

By, for instance, Lemma C1 of [1], if $k > 0$ is fixed and $d/n \to \infty$, then $E(\tilde{\lambda}_n^{-k}) = O(d^{-k})$. Thus,

$$
E_V\left(\frac{\tau^2}{l_1}\right)^k \leq E\left\{\frac{1}{(\psi^2 d+1)\|\mathbf{z}_1\|^2+\tilde{\lambda}_n}\right\}^k = O\left\{\left(\frac{1}{\psi^2 dn+d+n}\right)^k\right\},
$$

which proves (2). To prove (3), we use (4) and Lemma D1 below to obtain

$$
\begin{aligned}
E_V\left|\frac{\tau^2}{l_1} - \frac{1}{(\psi^2 d+1)\|\mathbf{z}_1\|^2+d}\right| &\leq E_V\left[\frac{\tau^2}{l_1\{(\psi^2 d+1)\|\mathbf{z}_1\|^2+d\}}\left|l_1 - (\psi^2 d+1)\|\mathbf{z}_1\|^2 - d\right|\right] \\
&= O\left\{\frac{d+n}{(\psi^2 dn+d+n)^2}\left(\frac{n}{d+n}\right)^{1/2}\right\}.
\end{aligned}
$$

$\square$

**Lemma A2.** *If $n \geq 9$ and $d \geq 1$, then*

$$
E_V\left\{(\mathbf{u}^T\hat{\mathbf{u}}_1)^2 - \frac{(\psi^2 d+1)\|\mathbf{z}_1\|^2}{(\psi^2 d+1)\|\mathbf{z}_1\|^2+d}\right\}^2 = O\left\{\left(\frac{d+n}{\psi^2 dn+d+n}\right)^2\frac{n}{d+n}\right\}.
$$

*Proof.* Throughout the proof, we let $c > 0$ be an absolute constant, which may vary from line to line, as necessary. Let $\hat{\mathbf{v}}_1, ..., \hat{\mathbf{v}}_n \in \mathbb{R}^n$ be unit-length eigenvectors of $XX^T$ corresponding to the eigenvalues $l_1 \geq \cdots \geq l_n \geq 0$. Observe that

$$
(\mathbf{u}^T\hat{\mathbf{u}}_1)^2 = \frac{\tau^2}{l_1}(\psi^2 d+1)(\mathbf{z}_1^T\hat{\mathbf{v}}_1)^2, \tag{5}
$$

The identity

$$(\psi^2 d + 1)||\mathbf{z}_1||^4 + \sum_{j=2}^{d}(\mathbf{z}_1^T \mathbf{z}_j)^2 = \frac{1}{\tau^2}\mathbf{z}_1^T X X^T \mathbf{z}_1 = \sum_{i=1}^{n}\frac{l_i}{\tau^2}(\mathbf{z}_1^T \mathbf{v}_i)^2$$

implies that

$$(\mathbf{z}_1^T \hat{\mathbf{v}}_1)^2 = \frac{\tau^2}{l_1}\left\{(\psi^2 d + 1)||\mathbf{z}_1||^4 + \sum_{j=2}^{d}(\mathbf{z}_1^T \mathbf{z}_j)^2 - \sum_{i=2}^{n}\frac{l_i}{\tau^2}(\mathbf{z}_1^T \hat{\mathbf{v}}_i)^2\right\},$$

and it follows that

$$(\mathbf{z}_1^T \hat{\mathbf{v}}_1)^2 \leq \frac{\tau^2}{l_1}\left\{(\psi^2 d + 1)||\mathbf{z}_1||^4 + \sum_{j=2}^{d}(\mathbf{z}_1^T \mathbf{z}_j)^2\right\}. \tag{6}$$

To find a good lower bound for $(\mathbf{z}_1^T \hat{\mathbf{v}}_1)^2$, observe

$$
\begin{aligned}
(\psi^2 d + 1)||\mathbf{z}_1||^4 + \sum_{j=2}^{d}(\mathbf{z}_1^T \mathbf{z}_j)^2 - \frac{l_2}{\tau^2}||\mathbf{z}_1||^2 \quad &\leq \quad \frac{l_1 - l_2}{\tau^2}(\mathbf{z}_1^T \hat{\mathbf{v}}_1)^2 \\
&= \quad \left\{\frac{l_1 - l_2}{\tau^2} - (\psi^2 d + 1)||\mathbf{z}_1||^2\right\}(\mathbf{z}_1^T \hat{\mathbf{v}}_1)^2 \\
&\quad + (\psi^2 d + 1)||\mathbf{z}_1||^2(\mathbf{z}_1^T \hat{\mathbf{v}}_1)^2 \\
&\leq \quad \left|\frac{l_1 - l_2}{\tau^2} - (\psi^2 d + 1)||\mathbf{z}_1||^2\right|||\mathbf{z}_1||^2 \\
&\quad + (\psi^2 d + 1)||\mathbf{z}_1||^2(\mathbf{z}_1^T \hat{\mathbf{v}}_1)^2.
\end{aligned}
$$

Thus,

$$
\begin{aligned}
(\mathbf{z}_1^T \hat{\mathbf{v}}_1)^2 \quad &\geq \quad \frac{1}{(\psi^2 d + 1)||\mathbf{z}_1||^2}\left\{(\psi^2 d + 1)||\mathbf{z}_1||^4 + \sum_{j=2}^{d}(\mathbf{z}_1^T \mathbf{z}_j)^2 - \frac{l_2}{\tau^2}||\mathbf{z}_1||^2\right\} \\
&\quad - \frac{1}{(\psi^2 d + 1)}\left|\frac{l_1 - l_2}{\tau^2} - (\psi^2 d + 1)||\mathbf{z}_1||^2\right|. \tag{7}
\end{aligned}
$$

Combining (5) with (6)-(7) yields

$$E_V\left\{(\mathbf{u}^T \hat{\mathbf{u}}_1)^2 - \frac{(\psi^2 d + 1)||\mathbf{z}_1||^2}{(\psi^2 d + 1)||\mathbf{z}_1||^2 + d}\right\}^2 \leq c(J_1 + J_2 + J_3 + J_4), \tag{8}$$

where

$$
\begin{aligned}
J_1 &= E_V\left[\frac{\tau^4}{l_1^2}(\psi^2 d + 1)\left\{(\psi^2 d + 1)||\mathbf{z}_1||^4 + \sum_{j=2}^{d}(\mathbf{z}_1^T \mathbf{z}_j)^2\right\} - \frac{\tau^2}{l_1}(\psi^2 d + 1)||\mathbf{z}_1||^2\right]^2, \\
J_2 &= E_V\left\{\frac{\tau^2}{l_1}(\psi^2 d + 1)||\mathbf{z}_1||^2 - \frac{(\psi^2 d + 1)||\mathbf{z}_1||^2}{(\psi^2 d + 1)||\mathbf{z}_1||^2 + d}\right\}^2, \\
J_3 &= E_V\left[\frac{\tau^2}{l_1}\left\{\sum_{j=2}^{d}\frac{(\mathbf{z}_1^T \mathbf{z}_j)^2}{||\mathbf{z}_1||^2} - \frac{l_2}{\tau^2}\right\}\right]^2, \\
J_4 &= E_V\left[\frac{\tau^2}{l_1}\left\{\frac{l_1 - l_2}{\tau^2} - (\psi^2 d + 1)||\mathbf{z}_1||^2\right\}\right]^2.
\end{aligned}
$$

To bound $J_1, J_2, J_3, J_4$, first observe that

$$(\psi^2 d + 1)||\mathbf{z}_1||^2 + \tilde{\lambda}_n \leq \frac{l_1}{\tau^2} \leq (\psi^2 d + 1)||\mathbf{z}_1||^2 + \tilde{\lambda}_1, \tag{9}$$

where $\tilde{\lambda}_j$ is the $j$-th largest eigenvalue of $\sum_{j=2}^d \mathbf{z}_j \mathbf{z}_j^T$. Then, by (9),

$$
\begin{aligned}
J_1 &= E_V \left[ \frac{\tau^4(\psi^2 d + 1)}{l_1^2} \left\{ (\psi^2 d + 1)||\mathbf{z}_1||^4 + \sum_{j=2}^d (\mathbf{z}_1^T \mathbf{z}_j)^2 - ||\mathbf{z}_1||^2 l_1 \right\} \right]^2 \\
&\leq (\psi^2 d + 1)^2 \left\{ E_V \left( \frac{\tau^2}{l_1} \right)^{13/3} \right\}^{12/13} \\
&\qquad \cdot \left[ E_V \left\{ (\psi^2 d + 1)||\mathbf{z}_1||^4 + \sum_{j=2}^d (\mathbf{z}_1^T \mathbf{z}_j)^2 - ||\mathbf{z}_1||^2 l_1 \right\}^{26} \right]^{1/13} \\
&\leq (\psi^2 d + 1)^2 \left\{ E \left( \frac{\tau^2}{l_1} \right)^{13/3} \right\}^{12/13} \\
&\qquad \cdot \left[ E \left\{ \sum_{j=2}^d (\mathbf{z}_1^T \mathbf{z}_j)^2 - ||\mathbf{z}_1||^2 \tilde{\lambda}_1 \right\}^{26} + E \left\{ \sum_{j=2}^d (\mathbf{z}_1^T \mathbf{z}_j)^2 - ||\mathbf{z}_1||^2 \tilde{\lambda}_n \right\}^{26} \right]^{1/13}.
\end{aligned}
$$

Let $W_{d-1} \sim \chi_{d-1}^2$ be a chi-squared random variable with $d-1$ degrees of freedom. By Lemma A1,

$$
E_V \left( \frac{\tau^2}{l_1} \right)^{13/3} = O \left\{ \left( \frac{1}{\psi^2 dn + d + n} \right)^{13/3} \right\}
$$

and by Lemma D1,

$$
\begin{aligned}
E \left\{ \sum_{j=2}^d (\mathbf{z}_1^T \mathbf{z}_j)^2 - ||\mathbf{z}_1||^2 \tilde{\lambda}_k \right\}^{26} &\leq cE \left\{ ||\mathbf{z}_1||^{52} (W_{d-1} - d + 1)^{26} \right\} \\
&\qquad + cE \left\{ ||\mathbf{z}_1||^{52} (\tilde{\lambda}_j - d + 1)^{26} \right\} \\
&= O \left( d^{13} n^{39} + n^{52} \right)
\end{aligned}
$$

for $k = 1, m$. It follows that

$$
J_1 = O \left\{ \frac{(\psi^2 d + 1)^2 (dn^3 + n^4)}{(\psi^2 dn + d + n)^4} \right\} = O \left\{ \left( \frac{d + n}{\psi^2 dn + d + n} \right)^2 \frac{n}{d + n} \right\}. \tag{10}
$$

To bound $J_2$, we have

$$
\begin{aligned}
J_2 &= E_V \left[ \frac{(\psi^2 d + 1)||\mathbf{z}_1||^2 \left\{ (\psi^2 d + 1)||\mathbf{z}_1||^2 + d - l_1 \right\}}{l_1 \{ (\psi^2 d + 1)||\mathbf{z}_1||^2 + d \}} \right]^2 \\
&\leq E_V \left[ \frac{(\psi^2 d + 1)||\mathbf{z}_1||^2 (\tilde{\lambda}_1 - d)}{l_1 \{ (\psi^2 d + 1)||\mathbf{z}_1||^2 + d \}} \right]^2 + E_V \left[ \frac{(\psi^2 d + 1)||\mathbf{z}_1||^2 (\tilde{\lambda}_n - d)}{l_1 \{ (\psi^2 d + 1)||\mathbf{z}_1||^2 + d \}} \right]^2 \\
&= O \left\{ \frac{(\psi^2 d + 1)^2 (dn^3 + n^4)}{(\psi^2 dn + d + n)^4} \right\} \\
&= O \left\{ \left( \frac{d + n}{\psi^2 dn + d + n} \right)^2 \frac{n}{d + n} \right\}. \tag{11}
\end{aligned}
$$

where the first inequality above follows by (9) and the subsequent equality follows as in our analysis of $J_1$ above.

The terms $J_2$ and $J_3$ involve $l_2$, the second largest eigenvalue of $XX^T$. A basic bound on $l_2$, which follows from Weyl's inequality (see, for instance, [2]), is

$$
\tilde{\lambda}_n \leq \frac{l_2}{\tau^2} \leq \tilde{\lambda}_1. \tag{12}
$$

Thus, using Lemma A1, (9), and (12), we have

$$J_3 \leq \left\{ E_V \left( \frac{\tau^2}{l_1} \right)^4 \right\}^{1/2} \left[ E(W_{d-1} - d + 1)^4 + E_V \left( \frac{l_2}{\tau^2} - d + 1 \right)^4 \right]^{1/2}$$

$$= O \left\{ \left( \frac{d+n}{\psi^2 dn + d + n} \right)^2 \frac{n}{d+n} \right\} \tag{13}$$

and

$$J_4 \leq \left\{ E_V \left( \frac{\tau^2}{l_1} \right)^4 \right\}^{1/2} \left\{ E(\tilde{\lambda}_1 - \tilde{\lambda}_n)^4 \right\}^{1/2} = O \left\{ \left( \frac{d+n}{\psi^2 dn + d + n} \right)^2 \frac{n}{d+n} \right\}. \tag{14}$$

The lemma follows by combining (8) with (10)-(11) and (13)-(14). $\qquad\square$

**Corollary A1.** *Suppose that $n \geq 9$ and $d \geq 1$. Then*

$$E_V(\mathbf{u}^T \hat{\mathbf{u}}_1)^2 = \frac{\psi^2 dn}{\psi^2 dn + d + n} + O \left\{ \frac{d+n}{\psi^2 dn + d + n} \left( \frac{n}{d+n} \right)^{1/2} + \frac{(\psi^2 dn)^2(d+n)}{(\psi^2 dn + d + n)^3} \frac{1}{n} \right\}$$

*Proof.* By Lemma A2,

$$E_V(\mathbf{u}^T \hat{\mathbf{u}}_1)^2 = \frac{(\psi^2 d + 1)||\mathbf{z}_1||^2}{(\psi^2 d + 1)||\mathbf{z}_1||^2 + d} + O \left\{ \frac{d+n}{\psi^2 dn + d + n} \left( \frac{n}{d+n} \right)^{1/2} \right\}.$$

The corollary follows since a basic Taylor expansion implies

$$\frac{(\psi^2 d + 1)||\mathbf{z}_1||^2}{(\psi^2 d + 1)||\mathbf{z}_1||^2 + d} = \frac{\psi^2 dn}{\psi^2 dn + d + n}$$

$$+ O \left\{ \frac{d+n}{\psi^2 dn + d + n} \left( \frac{n}{d+n} \right)^{1/2} + \frac{(\psi^2 dn)^2(d+n)}{(\psi^2 dn + d + n)^3} \frac{1}{n} \right\}.$$

$\qquad\square$

**Lemma A3.** *If $n \geq 9$ and $d \geq 1$, then*

$$E_V \left\{ \frac{\tau^2}{l_1} (\mathbf{u}^T \hat{\mathbf{u}}_1)^2 \right\} = E \left[ \frac{(\psi^2 d + 1)||\mathbf{z}_1||^2}{\{(\psi^2 d + 1)||\mathbf{z}_1||^2 + d\}^2} \right] + O \left\{ \frac{d+n}{(\psi^2 dn + d + n)^2} \left( \frac{n}{d+n} \right)^{1/2} \right\}$$

$$= \frac{\psi^2 dn}{(\psi^2 dn + d + n)^2} + O \left\{ \frac{d+n}{(\psi^2 dn + d + n)^2} \left( \frac{n}{d+n} \right)^{1/2} \right\}$$

$$+ O \left\{ \frac{(\psi^2 dn)^2}{(\psi^2 dn + d + n)^3} \frac{1}{n} \right\}.$$

*Proof.* By (5)-(7),

$$E_V \left| \frac{\tau^2}{l_1} (\mathbf{u}^T \hat{\mathbf{u}}_1)^2 - \frac{(\psi^2 d + 1)||\mathbf{z}_1||^2}{\{(\psi^2 d + 1)||\mathbf{z}_1||^2 + d\}^2} \right| \leq c(J_1 + J_2 + J_3 + J_4),$$

where $c > 0$ is an absolute constant and

$$J_1 = E_V \left| \frac{\tau^6}{l_1^3} (\psi^2 d + 1) \left\{ (\psi^2 d + 1)||\mathbf{z}_1||^4 + \sum_{j=2}^d (\mathbf{z}_1^T \mathbf{z}_j)^2 \right\} - \frac{\tau^4}{l_1^2} (\psi^2 d + 1)||\mathbf{z}_1|||^2 \right|,$$

$$J_2 = E_V \left| \frac{\tau^4}{l_1^2} (\psi^2 d + 1)||\mathbf{z}_1||^2 - \frac{(\psi^2 d + 1)||\mathbf{z}_1||^2}{\{(\psi^2 d + 1)||\mathbf{z}_1||^2 + d\}^2} \right|,$$

$$J_3 = E_V \left| \frac{\tau^4}{l_1^2} \left\{ \sum_{j=2}^d \frac{(\mathbf{z}_1^T \mathbf{z}_j)^2}{||\mathbf{z}_1||^2} - \frac{l_2}{\tau^2} \right\} \right|,$$

$$J_4 = E_V \left| \frac{\tau^4}{l_1^2} \left\{ \frac{l_1 - l_2}{\tau^2} - (\psi^2 d + 1)||\mathbf{z}_1||^2 \right\} \right|.$$

Bounds on $J_1, J_2, J_3, J_4$ follow as in the proof Lemma A2 and imply that

$$E_V \left| \frac{\tau^2}{l_1}(\mathbf{u}^T \hat{\mathbf{u}}_1)^2 - \frac{(\psi^2 d + 1)||\mathbf{z}_1||^2}{\{(\psi^2 d + 1)||\mathbf{z}_1||^2 + d\}^2} \right| = O \left\{ \frac{d + n}{(\psi^2 dn + d + n)^2} \left( \frac{n}{d + n} \right)^{1/2} \right\}.$$

The first equality in the lemma follows; the second equality follows from a basic Taylor expansion applied to

$$\frac{(\psi^2 d + 1)||\mathbf{z}_1||^2}{\{(\psi^2 d + 1)||\mathbf{z}_1||^2 + d\}^2}.$$

$\square$

## Appendix B: Proof of Theorem 2 (b)

Similar to the decomposition (1), we have the following decomposition for the risk of $\hat{y}_{bc}$:

$$R_V(\hat{y}_{bc}) = \sum_{j=1}^{9} I_j, \tag{15}$$

where

$$
\begin{aligned}
I_1 &= \sigma^2, \\
I_2 &= \sigma^2 E_V \left\{ \frac{\tau^2 l_1}{(l_1 - l_n)^2} \right\}, \\
I_3 &= \sigma^2 \psi^2 d E_V \left\{ \frac{\tau^2 l_1}{(l_1 - l_n)^2}(\mathbf{u}^T \hat{\mathbf{u}}_1)^2 \right\}, \\
I_4 &= \theta^2 \eta^2 \left( \frac{\psi^2 d}{\psi^2 d + 1} \right)^2 E_V \left\{ \frac{l_1}{l_1 - l_n}(\mathbf{u}^T \hat{\mathbf{u}}_1)^2 - 1 \right\}^2, \\
I_5 &= \frac{\theta^2 \eta^2}{(\psi^2 d + 1)^2}, \\
I_6 &= -2\theta^2 \eta^2 \frac{\psi^2 d}{(\psi^2 d + 1)^2} E_V \left\{ \frac{l_1}{l_1 - l_n}(\mathbf{u}^T \hat{\mathbf{u}}_1)^2 - 1 \right\}, \\
I_7 &= \frac{\theta^2 \eta^2}{\psi^2 d + 1} E_V \left\{ \frac{\tau^2 l_1}{(l_1 - l_n)^2} \right\}, \\
I_8 &= \theta^2 \eta^2 \frac{\psi^2 d}{\psi^2 d + 1} E_V \left\{ \frac{\tau^2 l_1}{(l_1 - l_n)^2}(\mathbf{u}^T \hat{\mathbf{u}}_1)^2 \right\}, \\
I_9 &= \theta^2 \eta^2 \frac{\psi^2 d}{(\psi^2 d + 1)^2} E_V \left( \frac{l_1}{l_1 - l_n}\mathbf{u}^T \hat{\mathbf{u}}_1 \right)^2.
\end{aligned}
$$

Lemmas B1-B3 below imply that

$$
\begin{aligned}
I_2 &= O \left\{ \frac{\sigma^2}{d} \frac{\psi^2 n + 1}{(\psi^2 n + \sqrt{n/d})^2} \right\}, \\
I_3 &= \sigma^2 \psi^2 E_V \left( \frac{1}{\psi^2 ||\mathbf{z}_1||^2 + \sqrt{n/d}} \right) + O \left\{ \sigma^2 \sqrt{\frac{n}{d}} \frac{\psi^2}{(\psi^2 n + \sqrt{n/d})^2} \right\}, \\
I_6 &= O \left\{ \frac{\theta^2 \eta^2}{\psi^2 d + 1} \sqrt{\frac{n}{d}} \left( \frac{1}{\psi^2 n + \sqrt{n/d}} \right) \right\}, \\
I_7 &= O \left\{ \frac{\theta^2 \eta^2}{d(\psi^2 d + 1)} \frac{\psi^2 n + 1}{(\psi^2 n + \sqrt{n/d})^2} \right\},
\end{aligned}
$$

$$I_8 = \frac{\theta^2 \eta^2}{\psi^2 d + 1} E\left(\frac{\psi^2}{\psi^2 ||\mathbf{z}_1||^2 + \sqrt{n/d}}\right) + O\left\{\frac{\theta^2 \eta^2}{\psi^2 d + 1} \sqrt{\frac{n}{d}} \frac{\psi^2}{(\psi^2 n + \sqrt{n/d})^2}\right\},$$

$$I_9 = \frac{\theta^2 \eta^2}{\psi^2 d + 1} E\left(\frac{\psi^2 ||\mathbf{z}_1||^2 + 1}{\psi^2 |||z_1||^2 + \sqrt{n/d}}\right) + O\left\{\frac{\theta^2 \eta^2}{\psi^2 d + 1} \sqrt{\frac{n}{d}} \frac{\psi^2 n + 1}{(\psi^2 n + \sqrt{n/d})^2}\right\}.$$

Theorem 2 (b) follows by combining these approximations with (15).

The rest of this appendix is devoted to the statement of proof of Lemma B1-B3.

**Lemma B1.** *Suppose that $k > 0$ is a fixed positive number. If $d \geq n > \max\{2k, k+3\}$, then*

$$E_V\left(\frac{\tau^2}{l_1 - l_n}\right)^k = O\left\{\left(\frac{1}{\psi^2 dn + \sqrt{dn}}\right)^k\right\}.$$

*Proof.* The first step in the proof is find lower bounds for $(l_1 - l_n)/\tau^2$. Let $\mathbf{q}_1 = \mathbf{z}_1/||\mathbf{z}_1||$ and let $\mathbf{q}_2 \in \mathbb{R}^n$ be a random unit vector that is orthogonal to $\mathbf{q}_1$. By projecting $ZZ^T$ into the two-dimensional subspace spanned by $\mathbf{q}_1, \mathbf{q}_2$, we find

$$\frac{l_1 - l_n}{\tau^2} \geq \sqrt{\{(\psi^2 d + 1)||\mathbf{z}_1||^2 + ||\mathbf{w}_1||^2 + ||\mathbf{w}_2||^2\}^2 + 4(\mathbf{w}_1^T \mathbf{w}_2)^2}, \qquad (16)$$

where $\mathbf{w}_1, \mathbf{w}_2 \sim N(0, I)$ are $d$-dimensional normal random vectors that are independent of each other and independent of $\mathbf{z}_1$.

On the other hand, by considering the orthogonal complement of the subspace spanned by $\mathbf{q}_1, \mathbf{q}_2$, it follows that there is an $(n-2) \times (d-1)$ matrix $\check{Z}$ with iid $N(0,1)$ entries such that $l_1/\tau^2 \geq \check{\lambda}_1$ and $l_n/\tau^2 \leq \check{\lambda}_n$, where $\check{\lambda}_1$ and $\check{\lambda}_{n-2}$ are the largest and smallest eigenvalues of $\check{Z}\check{Z}^T$, respectively. Thus,

$$\frac{l_1 - l_n}{\tau^2} \geq \check{\lambda}_1 - \check{\lambda}_{n-2}. \qquad (17)$$

Furthermore, $\check{\lambda}_1, \check{\lambda}_{n-2}$ are independent of $\mathbf{z}_1, \mathbf{w}_1, \mathbf{w}_2$.

Fix $0 < r < 1$. By (16)-(17),

$$E_V\left(\frac{\tau^2}{l_1 - l_n}\right)^k \leq J_1 + J_2, \qquad (18)$$

where

$$J_1 = \left(\frac{1}{1-r}\right)^k E\left\{\frac{1}{(\psi^2 d + 1)||\mathbf{z}_1||^2}\right\}^k,$$

$$J_2 = E\left\{\left(\frac{1}{\check{\lambda}_1 - \check{\lambda}_{n-2}}\right)^k ; ||\mathbf{w}_1||^2 - ||\mathbf{w}_2||^2 < -r(\psi^2 d + 1)||\mathbf{z}_1||^2\right\}.$$

Clearly,

$$J_1 = O\left\{\left(\frac{1}{\psi^2 dn + n}\right)^k\right\}. \qquad (19)$$

By independence, Markov's inequality, and Lemma D2,

$$\begin{aligned}
J_2 &= E\left(\frac{1}{\check{\lambda}_1 - \check{\lambda}_{n-2}}\right)^k P\left\{||\mathbf{w}_1||^2 - ||\mathbf{w}_2||^2 < -r(\psi^2 d + 1)||\mathbf{z}_1||^2\right\} \\
&\leq E\left(\frac{1}{\check{\lambda}_1 - \check{\lambda}_{n-2}}\right)^k E\left|\frac{||\mathbf{w}_1||^2 - ||\mathbf{w}_2||^2}{r(\psi^2 d + 1)||\mathbf{z}_1||^2}\right|^k \\
&= O\left[\left\{\frac{1}{\sqrt{n}(\psi^2 dn + n)}\right\}^k\right]. \qquad (20)
\end{aligned}$$

Combining (18)-(20) and the bound

$$E_V\left(\frac{\tau^2}{l_1 - l_n}\right)^k \leq E\left(\frac{1}{\check{\lambda}_1 - \check{\lambda}_{n-2}}\right)^k = O\{(nd)^{-k/2}\},$$

which follows from (17) and Lemma D2, implies

$$E_V \left( \frac{\tau^2}{l_1 - l_n} \right)^k = O \left\{ \left( \frac{1}{\psi^2 dn + \sqrt{dn}} \right)^k \right\}.$$

This completes the proof of the lemma. □

**Lemma B2.** *If $d \geq n \geq 7$, then*

$$E_V \left\{ \frac{\tau^2 l_1}{(l_1 - l_n)^2} \right\} = E \left\{ \frac{\psi^2 d ||\mathbf{z}_1||^2 + d}{(\psi^2 d ||\mathbf{z}_1||^2 + \sqrt{dn})^2} \right\} + O \left\{ \frac{(\psi^2 dn + d)\sqrt{dn}}{(\psi^2 dn + \sqrt{dn})^3} \right\}.$$

*Proof.* By Taylor's theorem,

$$\frac{\tau^2 l_1}{(l_1 - l_n)^2} - \frac{l_1}{\tau^2 (\psi^2 d ||\mathbf{z}_1||^2 + \sqrt{dn})^2} = -\frac{2l_1}{\tau^2 w_0^3} \left( \frac{l_1 - l_n}{\tau^2} - \psi^2 d ||\mathbf{z}_1||^2 - \sqrt{dn} \right), \qquad (21)$$

where $w_0$ lies in between $(l_1 - l_n)/\tau^2$ and $\psi^2 d ||\mathbf{z}_1||^2 + \sqrt{dn}$. It is clear that

$$E_V \left| \frac{l_1 \{ (l_1 - l_n)/\tau^2 - \psi^2 d ||\mathbf{z}_1||^2 - \sqrt{dn} \}}{\tau^2 (\psi^2 d ||\mathbf{z}_1||^2 + \sqrt{dn})^3} \right| = O \left\{ \frac{(\psi^2 dn + d)\sqrt{dn}}{(\psi^2 dn + \sqrt{dn})^3} \right\}. \qquad (22)$$

Additionally, by Hölder's inequality, (9), Lemma D1, and Lemma B1,

$$E_V \left| \frac{\tau^4 l_1 \{ (l_1 - l_n)/\tau^2 - \psi^2 d ||\mathbf{z}_1||^2 - \sqrt{dn} \}}{(l_1 - l_2)^3} \right|$$
$$\leq \left[ E_V \left\{ \frac{l_1}{\tau^2} \left( \frac{l_1 - l_n}{\tau^2} - \psi^2 d ||\mathbf{z}_1||^2 - \sqrt{dn} \right) \right\}^{10} \right]^{1/10}$$
$$\cdot \left\{ E_V \left( \frac{\tau^2}{l_1 - l_n} \right)^{30/9} \right\}^{9/10} \qquad (23)$$
$$= O \left\{ \frac{(\psi^2 dn + d)\sqrt{dn}}{(\psi^2 dn + \sqrt{dn})^3} \right\}.$$

Making use of (9) and Lemma D1 again, it follows from (21)-(23) that

$$E_V \left\{ \frac{\tau^2 l_1}{(l_1 - l_n)^2} \right\} = E \left\{ \frac{l_1}{\tau^2 (\psi^2 d ||\mathbf{z}_1||^2 + \sqrt{dn})^2} \right\} + O \left\{ \frac{(\psi^2 dn + d)\sqrt{dn}}{(\psi^2 dn + \sqrt{dn})^3} \right\}$$
$$= E \left\{ \frac{\psi^2 d ||\mathbf{z}_1||^2 + d}{(\psi^2 d ||\mathbf{z}_1||^2 + \sqrt{dn})^2} \right\} + O \left\{ \frac{(\psi^2 dn + d)\sqrt{dn}}{(\psi^2 dn + \sqrt{dn})^3} \right\},$$

which completes the proof of the lemma. □

**Lemma B3.** (a) *If $d \geq n \geq 7$, then*

$$E_V \left\{ \frac{\tau^2 l_1}{(l_1 - l_n)^2} (\mathbf{u}^T \hat{\mathbf{u}}_1)^2 \right\} = E \left( \frac{1}{\psi^2 d ||\mathbf{z}_1||^2 + \sqrt{dn}} \right) + O \left\{ \frac{\sqrt{dn}}{(\psi^2 dn + \sqrt{dn})^2} \right\}.$$

(b) *If $d \geq n \geq 7$, then*

$$E_V \left\{ \frac{l_1^2}{(l_1 - l_n)^2} (\mathbf{u}^T \hat{\mathbf{u}}_1)^2 \right\} = E \left\{ \frac{\psi^2 d ||\mathbf{z}_1||^2 + d}{\psi^2 d ||\mathbf{z}_1||^2 + \sqrt{dn}} \right\} + O \left\{ \frac{(\psi^2 dn + d)\sqrt{dn}}{(\psi^2 dn + \sqrt{dn})^2} \right\}.$$

(c) *If $d \geq n \geq 9$, then*

$$E_V \left\{ \frac{l_1}{l_1 - l_n} (\mathbf{u}^T \hat{\mathbf{u}}_1)^2 - 1 \right\}^2 = O \left\{ \frac{dn}{(\psi^2 dn + \sqrt{dn})^2} \right\}.$$

*Proof.* The proof of each part is very similar to the proof of Lemma A2. To prove part (a), (5)-(7) imply that

$$E_V \left| \frac{\tau^2 l_1}{(l_1 - l_n)^2} (\mathbf{u}^T \hat{\mathbf{u}}_1)^2 - \frac{1}{\psi^2 d ||\mathbf{z}_1||^2 + \sqrt{dn}} \right| \leq c(J_1 + J_2 + J_3 + J_4),$$

where $c > 0$ is a constant and

$$J_1 = E_V \left| \frac{\tau^6(\psi^2 d + 1)}{l_1(l_1 - l_n)^2} \left\{ (\psi^2 d + 1)||\mathbf{z}_1||^4 + \sum_{j=2}^{d} (\mathbf{z}_1^T \mathbf{z}_j)^2 \right\} - \frac{\tau^4(\psi^2 d + 1)||\mathbf{z}_1||^2}{(l_1 - l_n)^2} \right|,$$

$$J_2 = E_V \left| \frac{\tau^4(\psi^2 d + 1)||\mathbf{z}_1||^2}{(l_1 - l_n)^2} - \frac{1}{\psi^2 d ||\mathbf{z}_1||^2 + \sqrt{dn}} \right|,$$

$$J_3 = E_V \left| \frac{\tau^4}{(l_1 - l_n)^2} \left\{ \sum_{j=2}^{d} \frac{(\mathbf{z}_1^T \mathbf{z}_j)^2}{||\mathbf{z}_1||^2} - \frac{l_2}{\tau^2} \right\} \right|,$$

$$J_4 = E_V \left| \frac{\tau^4}{(l_1 - l_n)^2} \left\{ \frac{l_1 - l_2}{\tau^2} - (\psi^2 d + 1)||\mathbf{z}_1||^2 \right\} \right|.$$

The bounds

$$J_1, J_2, J_3, J_4 = O \left\{ \frac{\sqrt{dn}}{(\psi^2 dn + \sqrt{dn})^2} \right\}.$$

follow as in the proof of Lemma A2-A3, while making use of Lemma B1 where necessary. Part (a) of the lemma follows.

The relevant decomposition for part (b) is

$$E_V \left| \frac{l_1^2}{(l_1 - l_n)^2} (\mathbf{u}^T \hat{\mathbf{u}}_1)^2 - \frac{\psi^2 d ||\mathbf{z}_1||^2 + d}{\psi^2 d ||\mathbf{z}_1||^2 + \sqrt{dn}} \right| \leq c(J_1 + J_2 + J_3 + J_4),$$

where $c > 0$ is a constant and now

$$J_1 = E_V \left| \frac{\tau^4(\psi^2 d + 1)}{(l_1 - l_n)^2} \left\{ (\psi^2 d + 1)||\mathbf{z}_1||^4 + \sum_{j=2}^{d} (\mathbf{z}_1^T \mathbf{z}_j)^2 \right\} - \frac{\tau^2 l_1(\psi^2 d + 1)||\mathbf{z}_1||^2}{(l_1 - l_n)^2} \right|,$$

$$J_2 = E_V \left| \frac{\tau^2 l_1(\psi^2 d + 1)||\mathbf{z}_1||^2}{(l_1 - l_n)^2} - \frac{\psi^2 d ||\mathbf{z}_1||^2 + d}{\psi^2 d ||\mathbf{z}_1||^2 + \sqrt{dn}} \right|,$$

$$J_3 = E_V \left| \frac{\tau^2 l_1}{(l_1 - l_n)^2} \left\{ \sum_{j=2}^{d} \frac{(\mathbf{z}_1^T \mathbf{z}_j)^2}{||\mathbf{z}_1||^2} - \frac{l_2}{\tau^2} \right\} \right|,$$

$$J_4 = E_V \left| \frac{\tau^2 l_1}{(l_1 - l_n)^2} \left\{ \frac{l_1 - l_2}{\tau^2} - (\psi^2 d + 1)||\mathbf{z}_1||^2 \right\} \right|.$$

One can check that

$$J_1, J_2, j_3, J_4 = O \left\{ \frac{(\psi^2 dn + d)\sqrt{dn}}{(\psi^2 dn + \sqrt{dn})^2} \right\}.$$

Part (b) follows.

Finally, for part (c),

$$E_V \left\{ \frac{l_1}{l_1 - l_n} (\mathbf{u}^T \hat{\mathbf{u}}_1)^2 - 1 \right\}^2 \leq c(J_1 + J_2 + J_3 + J_4),$$

where $c > 0$ is a constant and

$$J_1 = E_V \left[ \frac{\tau^4(\psi^2 d + 1)}{l_1(l_1 - l_n)} \left\{ (\psi^2 d + 1)||\mathbf{z}_1||^4 + \sum_{j=2}^{d} (\mathbf{z}_1^T \mathbf{z}_j)^2 \right\} - \frac{\tau^2(\psi^2 d + 1)||\mathbf{z}_1||^2}{l_1 - l_n} \right]^2,$$

$$J_2 = E_V \left\{ \frac{\tau^2(\psi^2 d + 1)||\mathbf{z}_1||^2}{l_1 - l_n} - 1 \right\}^2,$$

$$J_3 = E_V \left[ \frac{\tau^2}{l_1 - l_n} \left\{ \sum_{j=2}^{d} \frac{(\mathbf{z}_1^T \mathbf{z}_j)^2}{||\mathbf{z}_1||^2} - \frac{l_2}{\tau^2} \right\} \right]^2,$$

$$J_4 = E_V \left[ \frac{\tau^2}{l_1 - l_n} \left\{ \frac{l_1 - l_2}{\tau^2} - (\psi^2 d + 1)||\mathbf{z}_1||^2 \right\} \right]^2.$$

Similar to above, it is straightforward to check that

$$J_1, J_2, J_3, J_4 = O \left\{ \frac{dn}{(\psi^2 dn + \sqrt{dn})^2} \right\}.$$

Thus, part (c) follows, which completes the proof of the lemma. □

## Appendix C: Proof of Proposition 1

Proposition 1 follows immediately from Lemmas A2 and B3.

## Appendix D: Additional lemmas

**Lemma D1.** *Let $k > 0$ be a fixed positive number. Then*

$$E|\lambda_n - d|^k, \ E|\lambda_1 - d|^k = O\{(dn)^{k/2} + n^k\}.$$

*Proof.* Suppose that $d \geq n$ and let $\lambda$ denote $\lambda_1$ or $\lambda_n$. By the Slepian-Gordon lemma,

$$\sqrt{d} - \sqrt{n} \leq E\lambda^{1/2} \leq \sqrt{d} + \sqrt{n}$$

and, by concentration of measure,

$$P\left(\lambda^{1/2} \geq \sqrt{d} + \sqrt{n} + t\right), \ P\left(\lambda^{1/2} \leq \sqrt{d} - \sqrt{n} - t\right) \leq e^{-t^2/2} \qquad (24)$$

(see, for instance, [3]).

Now observe that

$$E|\lambda - d|^k = \int_0^\infty P\left\{|\lambda - d|^k > t\right\} \ dt = I_1 + I_2, \qquad (25)$$

where

$$I_1 = \int_0^{d^k} P\left(\lambda^{1/2} < \sqrt{d - t^{1/k}}\right) \ dt,$$

$$I_2 = \int_0^\infty P\left(\lambda^{1/2} > \sqrt{d + t^{1/k}}\right) \ dt.$$

By (24),

$$I_1 = \int_0^{\{d - (\sqrt{d} - \sqrt{n})^2\}^k} P\left(\lambda^{1/2} < \sqrt{d - t^{1/k}}\right) \ dt$$

$$+ \int_{\{d - (\sqrt{d} - \sqrt{n})^2\}^k}^{d^k} P\left(\lambda^{1/2} < \sqrt{d - t^{1/k}}\right) \ dt$$

$$= \int_{\{d - (\sqrt{d} - \sqrt{n})^2\}^k}^{d^k} P\left(\lambda^{1/2} < \sqrt{d - t^{1/k}}\right) \ dt + O\{(dn)^{k/2}\}$$

$$= 2k \int_0^{\sqrt{d} - \sqrt{n}} \{d - (\sqrt{d} - \sqrt{n} - t)^2\}^{k-1}(\sqrt{d} - \sqrt{n} - t)$$

$$\cdot P\left(\lambda^{1/2} < \sqrt{d} - \sqrt{n} - t\right) \, dt + O\{(dn)^{k/2}\}$$

$$= O\left[\int_0^\infty \{d - (\sqrt{d} - \sqrt{n})^2\}^{k-1}(\sqrt{d} - \sqrt{n})e^{-t^2/2} \, dt + (dn)^{k/2}\right]$$

$$= O\{(dn)^{k/2}\}. \tag{26}$$

Similarly,

$$I_2 = \int_0^{\{(\sqrt{d}+\sqrt{n})^2 - d\}^k} P\left(\lambda^{1/2} > \sqrt{d + t^{1/k}}\right) \, dt$$

$$+ \int_{\{(\sqrt{d}+\sqrt{n})^2 - d\}^k}^\infty P\left(\lambda^{1/2} > \sqrt{d + t^{1/k}}\right) \, dt$$

$$= \int_{\{(\sqrt{d}+\sqrt{n})^2 - d\}^k}^\infty P\left(\lambda^{1/2} > \sqrt{d + t^{1/k}}\right) \, dt + O\{(dn)^{k/2} + n^k\}$$

$$= 2k\int_0^\infty \{(\sqrt{d} + \sqrt{n} + t)^2 - d\}^{k-1}(\sqrt{d} + \sqrt{n} + t)P\left(\lambda^{1/2} > \sqrt{d} + \sqrt{n} + t\right) \, dt$$

$$+ O\{(dn)^{k/2} + n^k\}$$

$$= O\left[\int_0^\infty \{(\sqrt{d} + \sqrt{n})^2 - d\}^{k-1}(\sqrt{d} + \sqrt{n})e^{-t^2/2} \, dt + (dn)^{k/2}\right]$$

$$= O\{(dn)^{k/2} + n^k\}. \tag{27}$$

For $d \geq n$, the lemma follows by combining (26)-(27) with (25). If $d < n$, the result for $\lambda_n$ is trivial, because $\lambda_n = 0$; the result for $\lambda_1$ follows from the $d \geq n$ case because $\lambda_1$ is also the largest eigenvalue of $ZZ^T$. $\qquad\square$

**Lemma D2.** *Suppose that $d \geq n \geq 2$. Then*

$$E\left(\frac{1}{\lambda_1 - \lambda_n}\right)^{n-1} = O\left\{e^n(nd)^{-(n-1)/2}\right\}.$$

*Proof.* The joint density of $(\lambda_1, ..., \lambda_n)$ is given by

$$f(\lambda_1, ..., \lambda_n) = K_{d,n} \exp\left(-\frac{1}{2}\sum_{i=1}^n \lambda_i\right) \prod_{i=1}^n \lambda_i^{(d-n-1)/2} \prod_{1 \leq i < j \leq n} (\lambda_i - \lambda_j), \quad \lambda_1 \geq \cdots \geq \lambda_n \geq 0,$$

where

$$K_{d,n}^{-1} = \left(\frac{2^d}{\pi}\right)^{n/2} \prod_{i=1}^n \Gamma\left(\frac{d - i + 1}{2}\right)\Gamma\left(\frac{n - i + 1}{2}\right)$$

(see, for instance, Chapter 3 of [4]). Let $R = \{(\lambda_1, ..., \lambda_n) \in \mathbb{R}^n; \ 0 \leq \lambda_1 \leq \cdots \leq \lambda_n\}$. Then

$$E\left(\frac{1}{\lambda_1 - \lambda_n}\right)^{n-1} = \int_R \left(\frac{1}{\lambda_1 - \lambda_n}\right)^{n-1} f(\lambda_1, ..., \lambda_n) \, d\lambda_1 \cdots d\lambda_n$$

$$\leq K_{d,n} \int_R \exp\left(-\frac{1}{2}\sum_{i=1}^n \lambda_i\right) \prod_{i=1}^n \lambda_i^{(d-n-1)/2}$$

$$\cdot \prod_{2 \leq i < j \leq n} (\lambda_i - \lambda_j) \, d\lambda_1 \cdots d\lambda_n$$

$$\leq \frac{K_{d,n}}{K_{d-1,n-1}} \int_0^\infty \lambda^{(d-n-1)/2} e^{-\lambda/2} \, d\lambda$$

$$= \frac{2^{1-n}\sqrt{\pi}}{\Gamma(d/2)\Gamma(n/2)}\Gamma\left(\frac{d - n + 1}{2}\right).$$

By Stirling's approximation,

$$\frac{2^{1-n}\sqrt{\pi}}{\Gamma(d/2)\Gamma(n/2)}\Gamma\left(\frac{d - n + 1}{2}\right) = O\left\{e^n(dn)^{-(n-1)/2}\right\}.$$

Thus,

$$E\left(\frac{1}{\lambda_1 - \lambda_n}\right)^{n-1} = O\left\{e^n(dn)^{-(n-1)/2}\right\}.$$

$\square$