[Reviews · NeurIPS 2013]

Submitted by Assigned_Reviewer_3

This paper proposes a simple latent factor model for one-shot learning with continuous outputs where very few observations are available. Specifically, it derives risk approximations in an asymptotic regime where the number of training examples is fixed and the number of features in the X space diverges. Based on principal component regression (PCR) estimator, two estimators including the bias-corrected estimator and the so-called "oracle" estimator are proposed and the bounds for the risks of these estimators are derived. These bounds provide insights into the significance of various parameters relevant to one-shot learning.
The major contribution of this paper is the bounds derived for 3 estimators: the principal component regression estimator, the bias-corrected estimator and the oracle estimator which assumes the first principal component is known.
Summary: The bounds of risks of PCR estimator in the one-shot learning are derived. A bias-corrected estimator based on PCR estimator is proposed with better consistency property.

Submitted by Assigned_Reviewer_5

UPDATE: After discussion with the other reviewers, I've lowered my score a little. The simple estimator suggested by another reviewer does seem to perform surprisingly well in the idealized model studied in this paper. I interpret this to mean that the model is a bit unrealistic. However, the PCR estimator still seems interesting, and the authors' feedback has helped clarify that.

This paper studies a linear latent factor model, where one observes "examples" consisting of high-dimensional vectors x1, x2, ... in R^d, and one wants to predict "labels" consisting of scalars y1, y2, ... in R. Crucially, one is working in the "one-shot learning" regime, where the number of training examples n is small (say, n=2 or n=10), while the dimension d is large (say, d -> infinity). This paper considers a well-known method, principal component regression (PCR), and proves some somewhat surprising theoretical results: PCR is inconsistent, but a modified PCR estimator is weakly consistent; the modified estimator is obtained by "expanding" the PCR estimator, which is different from the usual "shrinkage" methods for high-dimensional data.

Quality: The paper appears to be correct, though I have not checked the calculations in the supplementary material. I do have one question, which is more conceptual: how would the methods in this paper, which use PCR with 1 component, compare to PCR with more components?

In section 3, line 117, the authors argue that it is natural to restrict attention to PCR with 1 component, because when one computes the expectation of the matrix X^T X, one finds that the interesting information (the vector u) is in the leading eigenvector. However, I am not quite convinced by this argument, because in the "one-shot learning" regime, where n << d, the matrix X^T X is not at all close to its expectation.

Moreover, in practice, people have reported cases where PCR runs into exactly this problem -- when one keeps only the first few principal components, the resulting subspace does not contain the desired solution (see, e.g., Hadi and Ling, Amer. Stat. 52(1), p.15, Feb. 1998). Such behavior may be ruled out by the assumptions made in this paper -- specifically, the assumption that the vectors x_i have strong "signal-to-noise" ratio. Still, this seems like a significant point to mention, to illustrate that rather strong assumptions are needed for one-shot learning.

Clarity: The paper is clear and well organized. Some minor suggestions: move the definition of the bias-corrected PCR estimator to section 3 (instead of burying it in the middle of section 4); in table 2, when n=4, in the "oracle" column, there may be a typo (0.43% should be 4.3%); in section 8, line 424, the matrix S should possibly be transposed.

Originality: The main novelty of the paper seems to be the "one-shot learning" setting, and the modified PCR estimator. The latent factor model studied in this paper (and in particular the assumption that various distributions are Gaussian) seem standard and unremarkable. Nonetheless, the proofs require significant work.

I am not familiar with some of the related work mentioned in the introduction (lines 52-55). It would be helpful if the authors stated in more detail how this paper differs from the existing work.

Significance: It is impressive that the method works quite well (at least on synthetic data) for specific small values of n, like n=2 and n=9. It is also impressive that the theoretical bounds are not far off from the actual behavior (though again, this is on synthetic data). Overall this seems like a solid theory paper, and the open questions look interesting.
Summary: This paper considers a well-studied method, principal components regression, in a challenging setting, one-shot learning. This seems like a solid theory paper.

Submitted by Assigned_Reviewer_7

Update:

The point of my simple method was to obtain a better understanding of the theoretical analysis. I ran the same experiments the authors ran from section 7, with d=500,5000,500000 and the simple method began to converge to the oracle method and at d=500 it certainly did not have an error greater than 500 (roughly 7). I ran all of the experiments with n=2. **edit** The focus of these experiments were on weak convergence as that was the theoretical analysis performed for n=2. For n=2 the simple method does perform very poorly for squared loss, but performs better for absolute loss. **end edit**

I believe that this one-shot idea is very interesting, but the bias correction has not been thoroughly explored. The experiments are show a lot of promise, which is why I remain borderline with this paper.

====

***
This paper aims to provide an analysis for principle component
regression in the setting where the feature vectors $x$. The authors
let $x = v + e$ where $e$ is some corruption of the nominal feature
vector $v$; and $v = a u$ where $a \sim N(0,\eta^2 \gamma^2 d)$ while
the observations $y = \theta/(\gamma \sqrt{d}) \langle v,u \rangle + \xi$. This
formulation is slightly different than the standard one because our
design vectors are noisy, which can pose challenges in identifying the
linear relationship between $x$ and $y$. Thus, using the top principle
components of $x$ is a standard method used in order to help
regularize the estimation. The paper is relevant to the ML
community. The key message of using a bias-corrected estimate of $y$
is interesting, but not necessarily new. Handling bias in regularized
methods is a common problem (cf. Regularization and variable selection
via the Elastic Net, Zou and Hastie, 2005). The authors present
theoretical analysis to justify their results. I find the paper
interesting; however I am not sure if the number of new results and
level of insights warrants acceptance.

***
The result is interesting, but lacks enough depth to provide a
convincing argument for its use. The authors present a very specific
setting for the design vectors: spiked covariance. The motivation of
the bias correction is interesting, but how do the authors expect it
should work under other models for $x$ and $y$? A simulation study of
these methods could prove useful in order to provide insights into the
behavior in the non-Gaussian setting. Furthermore, as the authors
admit, the signal of $x$ is very strong in the direction of the
regression vectors. It is conceivable that the signal $x$ is strong in
multiple directions (especially in the high-dimensional setting). How
would the bias-correction change when adapting to multiple PCA
factors?

***
Overall, the model being used to perform the analysis seems to simple
to shed serious light on the problem. For example, one can show risk
consistency in the simple setting of letting the largest principle
component simply be equal $x_1$. That is, only perform PCA one the first
example and regression on the second.

The authors go through a lot of analysis from fundamentals without
providing intuition for why the shrinkage occurs. In particular, the
authors should explain why least squares (which produces consistent
estimators) must be corrected in this context. Interestingly, the
simple estimator of just taking $x_1$ to be the principle component
and performing regression on the remaining examples performs
better. Thus, rather than a shrinkage correction, a more reasonable
solution seems to be to learn the PC vectors on one set of data and
regress on the other. Can the authors provide more intuition for why
they expect their proposed method to perform better in generic settings?
Summary: I find this idea interesting; however, the authors do not provide enough intuition for why their bias correction should work in other settings outside of their specific model.
Author Feedback

Author rebuttal: We thank the reviewers for their thoughtful comments on the paper. While all of the reviewers' comments have been valuable, we'd like to take this space-limited opportunity to respond to four specific comments (numbered below) made by the reviewers.

1. One of the reviewers suggested a very simple alternative methodology, where the principal component direction is taken to be x_1 (the first predictor vector) and the estimator \hat{\beta} is obtained by performing principal component regression in this direction with the remaining n-1 observations. The reviewer further suggested that this elementary estimator might out-perform the estimators discussed in the paper -- if true, this could substantially negate the impact of our results.

In order to investigate these questions, we ran a simulation analysis involving the method proposed by the reviewer and the other methods discussed in the paper. We used the same settings as in Section 7 of the paper. The table below contains the empirical prediction error of the various methods for d=500 and several values of n (based on 1000 datasets).

PCR BC OR X1
n=2 18.0748 7.6869 1.7507 595.8016
n=4 6.4769 0.9444 0.3514 7.1833
n=9 1.3757 0.3430 0.2612 3.9847
n=20 0.4466 0.2721 0.2403 4.1778

Here, PCR refers to the basic principal component regression estimator discussed in the paper; BC refers to the bias-corrected estimator; OR refers to the oracle estimator; and X1 is the method suggested by the reviewer. Note that BC substantially outperforms X1 in all settings (PCR and OR also outperform X1 in all settings). Results for d=5000 (not reported here) were similar -- BC substantially outperformed X1 in all settings.

To help explain these results, let X=(x_2,…,x_n)^T denote the matrix consisting of the predictors x_2,…,x_n and let y=(y_2,…,y_n) denote the corresponding responses. Then the estimator suggested by the reviewer has the form \hat{\beta} = (x_1^TX^Ty/||Xx_1||^2)x_1. Notice that for large d and fixed n, the denominator in \hat{\beta} satisfies

||Xx_1||^2 = O(dR + d^2W_1W_{n-1}),

where R is a bounded (in probability) non-negative random variable, and W_1, W_{n-1} are independent chi-squared random variables on 1 and n-1 degrees of freedom, respectively. The main point is that the risk of \hat{\beta} involves inverse moments of ||Xx_1||^2 and that instability of 1/W_1 (indeed, E(1/W_1) = \infty) contributes to instability of \hat{\beta}.

The simulation results and the above discussion suggest that \hat{\beta} may not be a viable alternative to the approaches discussed in the paper. On the other hand, other split sample approaches to the problem could potentially be effective, but it seems unclear how these approaches might compare to the methods proposed in the paper. We believe that this is an interesting topic for future research.

2. In the submitted paper, we consider a simple latent factor model with Gaussian data. One of the reviewers asks "Can the authors provide more intuition for why
they expect their proposed method to perform better in generic settings?" Our key theoretical results primarily depend on lower-order moments of the random variables involved and basic facts about the eigenvalues and eigenvectors of random matrices. Methods relying only on lower-order moments are frequently useful in generic settings. Additionally, a great deal of recent work has demonstrated the universality of many properties of eigenvalues and eigenvectors of high-dimensional random matrices (see, for instance, Vershynin (2012) "Introduction to the non-asymptotic analysis of random matrices" or Pillai and Yin (2013) "Universality of covariance matrices"). Taken together, these observations suggest that the proposed methods may perform quite well under more relaxed distributional assumptions. (Additionally, see 3 below for a discussion of a more general multi-factor model -- this is another more generic setting in which methods related to those proposed in this paper may be useful.)

3. One of the reviewers asks "How would the methods in this paper, which use PCR with 1 component, compare to PCR with more components?" The results in this paper may be extended to a more general k-factor model, which corresponds to PCR with more than 1 component. In the k-factor model, k > 1 components link the predictors and outcomes, and k components are utilized in the regression. In the extended journal version of this paper (currently in preparation), we show that the "usual" PCR with k components is inconsistent, when n is fixed and d \to \infty, but that a simple bias-corrected k-component PCR method is consistent. Similar to the single component model studied here, in the k-component model, the bias-corrected estimator involves "correcting" for the contribution of noise in the sample eigenvectors.

4. One of the reviewers notes that "In the 'one-shot learning' regime, where n << d, the matrix X^TX is not at all close to its expectation" and questions the implications of this fact for the proposed methods. This observation is in fact one of the key issues that necessitates bias-correction for consistent prediction in the one-shot regime: If X^TX were closer to its expectation (as in "large n, small d" asymptotics), then bias-correction would not be required for consistency. One of the major implications of our work is that the discrepancy between X^TX and its expectation (as manifest through noise in the sample eigenvectors and eigenvalues) can be quantified and corrected-for, in order to achieve effective prediction in the one-shot regime.